

# Internal-wave-induced dissipation rates in the Weddell Sea Bottom Water gravity current

Ole Pinner[1], Friederike Pollmann[2], Markus Janout[1], Gunnar Voet[3,4], and Torsten Kanzow[1,5]

[1]Alfred Wegener Institute, Helmholtz Centre for Polar and Marine Research (AWI), Am Handelshafen 12, 27570 Bremerhaven
[2]CEN - Center for Earth System Research and Sustainability, Universität Hamburg, Bundesstraße 53, 20146 Hamburg, Germany
[3]University of California, San Diego, 9500 Gilman Drive, La Jolla, USA
[4]Scripps Institution of Oceanography, 9500 Gilman Drive, La Jolla, USA
[5]Department of Physics and Electrical Engineering, University of Bremen, Otto-Hahn-Allee 1, 28359 Bremen, Germany

**Correspondence:** Ole Pinner (ole.pinner@awi.de)

**Abstract.** This study investigates the role of wave-induced turbulence in the dynamics of the Weddell Sea Bottom Water gravity current. The current transports dense water from its formation sites on the shelf to the deep sea and is a crucial component of the Southern Ocean overturning circulation. The analysis is based on velocity records from a mooring array deployed across the continental slope between January 2017 and January 2019 and salinity and temperature (CTD) profiles measured by various

ship expeditions. To quantify the importance of internal waves for entrainment into the gravity current along the continental slope, we employ three independent methods for estimating turbulence. First, we use a Thorpe scale approach to compute turbulence from density inversions in density profiles in order to calculate total, process-independent dissipation rate. Second, we apply the finestructure parameterization to estimate wave-induced mixing from vertical profiles. Third, we estimate wave energy levels from moored velocity time series and deduce turbulent kinetic energy dissipation rates by applying a formulation

that is at the heart of the finestructure parameterization. On this transect, turbulence is highest on the shelf break and decreases towards the deep sea, in line with a decreasing strength of wave-induced turbulence. We observe a 2-layer structure of the gravity current, a strongly turbulent about $60$–$80\,\mathrm{m}$ thick bottom layer and an upper, more quiescent interfacial layer. In the interfacial layer, internal waves induce an important part of the dissipation rate and therefore to entrainment of warmer upper water into the gravity current. A literature comparison with turbulence measurements up- and downstream of our study site

suggests that the question of which turbulent process is dominant may be dependent on the location along the Weddell Sea Bottom Water gravity current. On the shelf, trapped waves are most important, on the slope, we see the effect of breaking internal waves and in the basin, symmetric instabilities are identified as the main driver of turbulence.



## 1 Introduction

The global overturning or thermohaline circulation is closed through deep water formation at high latitudes, connecting surface and deep sea currents. Nearly half of the circulation's densest water mass, the Antarctic Bottom Water, originates in the Weddell Sea (Hellmer and Beckmann, 2001). This gives the Weddell Sea, a marginal sea in the Southern Ocean, a critical role in the global ocean and climate dynamics. On the continental shelves of the Weddell Sea, Dense Shelf Water forms beneath ice shelves and during sea ice formation (Foldvik et al., 2004) and flows as a gravity current into the deep sea (Llanillo et al., 2023). Steered

by the Coriolis force, the current follows the Antarctic continental shelf, as indicated in Fig. 1a. The water mass transported by this gravity current is referred to as Weddell Sea Bottom Water (WSBW) and is categorized from its neutral density (Jackett and McDougall, 1997): WSBW is defined to have a higher neutral density than $\gamma^n = 28.40\,\mathrm{kg\,m^{-3}}$ (Naveira Garabato et al., 2002b), and with that is too dense to leave the Weddell Sea but through the deepest passages (Naveira Garabato et al., 2002a). The majority of the water leaving the Weddell Sea to become Antarctic Bottom Water is provided by Weddell Sea Deep Water

(WSDW), which is categorized to have a neutral density $28.26\,\mathrm{kg\,m^{-3}} < \gamma^n < 28.40\,\mathrm{kg\,m^{-3}}$ (Naveira Garabato et al., 2002b). It is formed through mixing processes of Weddell Sea Bottom Water with ambient lighter waters (Nicholls et al., 2009).

The physical properties of Antarctic Bottom Water are subsequently in part determined by processes at the formation sites on the continental shelves, but also by entrainment of upper, less dense water into the WSBW gravity current during its passage down the continental slope (Legg et al., 2009). This entrainment of ambient water is a consequence of vertical mixing

by multiple turbulent processes. Investigating the role and nature of the small-scale processes involved in the entrainment is therefore essential for advancing our understanding of Antarctic Bottom Water formation (Silvano et al., 2023, Question 7). This further understanding is especially needed as Antarctic Bottom Water has been observed to warm and freshen (Purkey and Johnson, 2013) at increasing rates (Menezes et al., 2017; Johnson et al., 2019) and is hypothesized to be a potential tipping point in the global climate system (Lenton et al., 2008). Global and regional models cannot resolve the required scales of

vertical mixing for simulating realistic deep water formation (Legg et al., 2009) and have to rely on parameterizations (Heuzé, 2021). For the development and constraint of these parameterizations and to understand turbulent entrainment into dense bottom currents, an observation-based approach is necessary. Many turbulent processes found in gravity currents are driven by the kinetic energy of the gravity current itself, like shear instabilities in the interface to the ambient water, or friction and drag at the sea floor (Legg et al., 2009). But only considering this driving mechanism would leave out the ever-present external

energy source of internal waves. While turbulence driven by breaking internal waves is the most important mixing mechanism in the open ocean and accordingly discussed in many publications, the interaction of internal waves and gravity currents is only described in few publications (Seim and Fer, 2011; Nash et al., 2012, etc.), of which some consider only very idealized setups (Hogg et al., 2018; Tanimoto et al., 2021, 2022). We hence aim to evaluate the importance of wave-induced turbulence for the WSBW gravity current.

The Weddell Sea features strong tidal currents (Foldvik et al., 1990; Levine et al., 1997; Robertson et al., 1998), suggesting a vigorous internal wave field produced by their interaction with rough topography. Due to its remote and difficult to access location at high latitudes, the Weddell Sea is not included in observation-based global maps of internal wave energy (Water-





house et al., 2014; de Lavergne et al., 2019; Pollmann, 2020; Pollmann et al., 2023). Our research is based on the so-called Joinville transect across the Antarctic continental slope, which covers the pathway of the WSBW gravity current in the north-

western Weddell Sea before it exits into the deep sea. We use moored and shipboard observations of velocity and hydrography along this transect (Fig. 1b & Fig. 1c) to quantify turbulence dependent on its driving energy source from three independent methods: 1.) a Thorpe scale approach, applied to CTD profiles, 2.) a parameterization based on the energy contained in internal waves, calculated from velocity time series, and 3.) strain-based finestructure parameterization from CTD profiles. We obtain the contribution of wave-induced turbulence to the overall turbulence by horizontal and vertical comparison of the results from

all three methods. From this, we can assess the relevance of internal waves for entrainment of ambient water into the WSBW gravity current.

## 2  Data

This study is based on multiple observations along a transect across the continental slope east of the Antarctic Peninsula (Fig. 1). This section briefly describes velocity mooring data and hydrographic CTD profiles along the transect. Detailed information

can be found in the respective cruise reports. We use salinity and temperature data from 13 RV *Polarstern* expeditions between 1989 and 2022 that collected CTD profiles along the Joinville transect (Table 1). Profiles from other RV *Polarstern* expeditions to the same region are rejected, as they measured along a different transect, offset from the one we consider. This is done to keep the CTD profiles spatially comparable in the cross-section along the continental slope. Profiles are depth-binned at 1 or 2 dbar resolution following standard procedures to reduce measurement noise and errors due to ship movement. During the

expedition PS129, the CTD was also equipped with LADCPs.

Seven moorings were deployed along the transect during the RV *Polarstern* expedition PS103 around New Year 2016/2017 and recovered in January 2019, during PS117. Horizontal spacing between the moorings is 35 to 50 km. Vertical resolution of the moored measurements ranges from 50 to 200 m (Fig. 1c). Results are presented in height above bottom, because the difference in total depth from the shelf sea to the deep sea is more than an order of magnitude larger than the height of the

bottom current of approximately 300 m. The moorings were equipped with three Aanderaa current meters (models RCM7, RCM8 and RCM11), three Seabird MicroCAT CTD sensors (SBE37) and 3 Seabird temperature-depth recorders (SBE39/56) each. The RCMs had an accuracy of $\pm 1\,\mathrm{cm\,s^{-1}}$ for speed and $\pm 5^{\circ}$ for direction and integrated vector velocity over 2 hour periods. Most velocity time series considered here are 2 years in total length, except for a few shorter records because of battery failure.

## 3  Methods

Water masses in the Weddell Sea are categorized from their neutral density $\gamma^{\mathrm{n}}$ (Naveira Garabato et al., 2002b). Therefore, we calculate neutral densities for each CTD profile with the MATLAB toolbox associated with Jackett and McDougall (1997). Results are averaged arithmetically in $0.5^{\circ}$ longitude bins to form mean background densities and are then used to differentiate





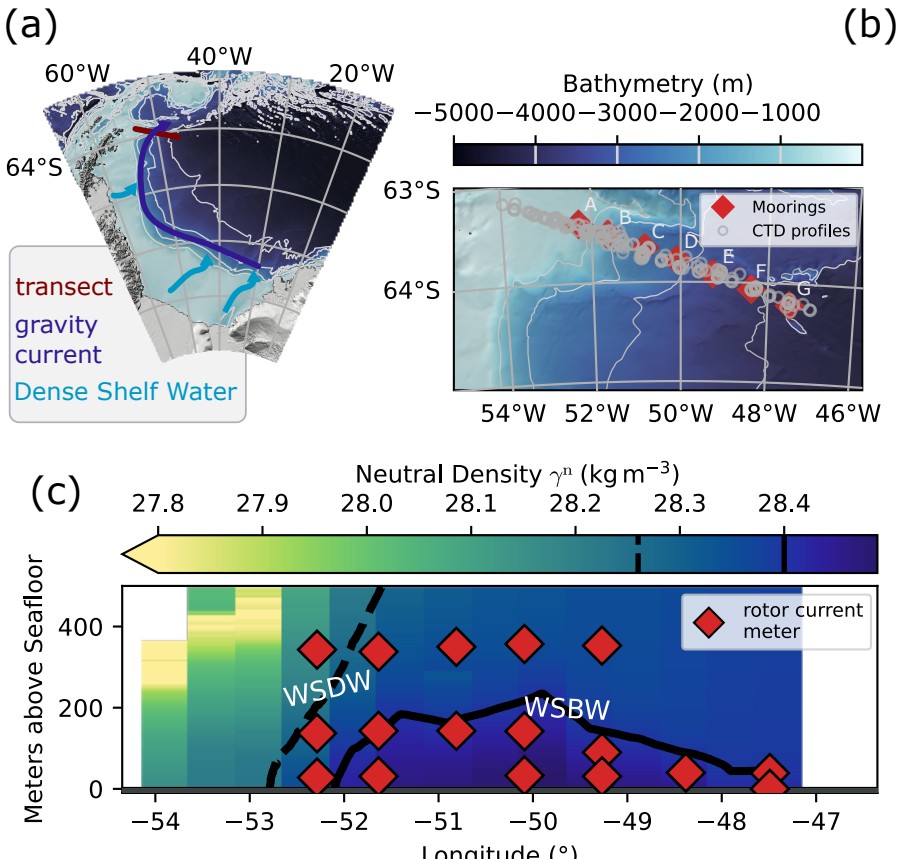

**Figure 1. (a)** Map of the Weddell Sea, with the Joinville transect across the continental slope in red. Light blue arrows show the path of Dense Shelf Water, which feed the Weddell Sea bottom water gravity current, shown in dark blue. Faint grey lines are isobaths in steps of 1000 m. **(b)** Map of the Joinville transect across the continental slope. Red diamonds mark positions of the 7 moorings, and grey dots CTD profiles. Moorings are named **A** to **G** from west to east, according to Table 2. Faint grey lines are isobaths in steps of 1000 m. **(c)** Neutral density $\gamma^n$ transect across the continental slope, with the Antarctic Peninsula to the west. Black lines denote water mass boundaries, derived from neutral density. The definition of Weddell Sea Bottom Water (WSBW) is also taken as the extent of the gravity current. Red diamonds show rotor current meter locations.

between water masses. Gravity current mean flow is calculated by taking long-time averages over the complete measurement

period for each complex velocity time series. The produced data points are first vertical and then horizontal linearly interpolated to yield an approximate mean flow field.

Our main goal is to quantify turbulence dependent on its driving energy source in the gravity current. The amount of turbulence is quantified by the dissipation rate $\varepsilon$ in units of $\mathrm{W\,kg^{-1}}$, the conversion rate of turbulent kinetic energy to heat. To do so, we apply three different methods. We first estimate the total turbulent kinetic energy dissipation by applying the Thorpe



| Cruise | Year | Citation | # profiles | Data type |
|---|---|---|---|---|
| ANT-VIII/2 | 1989 | Fahrbach and Rohardt (1990) | 10 | CTD |
| ANT-IX/2 | 1990/91 | Fahrbach and Rohardt (1991) | 10 | CTD |
| ANT-X/7 | 1992/93 | Fahrbach and Rohardt (1993) | 10 | CTD |
| ANT-XIII/4 | 1996 | Fahrbach and Rohardt (1996) | 8 | CTD |
| ANT-XV/4 | 1998 | Fahrbach and Rohardt (1998) | 20 | CTD |
| ANT-XXII/3 | 2005 | Rohardt (2010) | 14 | CTD |
| ANT-XXIV/3 | 2008 | Fahrbach and Rohardt (2008) | 23 | CTD |
| ANT-XXVII/2 | 2010/11 | Rohardt et al. (2011) | 13 | CTD |
| ANT-XXIX/2 | 2012/13 | Rohardt (2013) | 10 | CTD |
| ANT-XXIX/6 | 2013 | Lemke et al. (2013) | 7 | CTD |
| PS103 | 2016 | Rohardt and Boebel (2017) | 15 | CTD |
| PS117 | 2018 | Rohardt et al. (2022) | 10 | CTD |
| PS129 | 2022 | *in prep.* | 18 | CTD, LADCP |
| | | | 178 | |

**Table 1.** Ship-based data from RV *Polarstern* cruises along the transect between 1989 and 2022.

| Name | Latitude (°S) | Longitude (°W) | Water depth (m) | ID | Citation |
|---|---|---|---|---|---|
| A | 63.40 | 52.29 | 643 | AWI262-1 | Rohardt and Boebel (2019a) |
| B | 63.51 | 51.64 | 1656 | AWI261-1 | Rohardt and Boebel (2019b) |
| C | 63.66 | 50.81 | 2493 | AWI207-10 | Rohardt and Boebel (2019c) |
| D | 63.78 | 50.09 | 2757 | AWI260-1 | Rohardt and Boebel (2019d) |
| E | 63.92 | 49.27 | 3390 | AWI259-1 | Rohardt and Boebel (2019e) |
| F | 64.07 | 48.38 | 3876 | AWI258-1 | Rohardt and Boebel (2019f) |
| G | 64.22 | 47.49 | 4160 | AWI257-1 | Rohardt and Boebel (2019g) |

**Table 2.** Moorings along the Joinville transect from January 2017 to January 2019, along with their coordinates, total water depth, official ID, and reference. From the referenced data sets we use current velocity, time, and depth.

scale method to density profiles. Because the Thorpe scale approach does not distinguish between overturns produced by breaking internal waves, instabilities or other sources, it gives an estimation of the total dissipation rate. We then calculate the internal wave-induced turbulence with methods based on evaluations of spectral energy transfers by wave–wave interactions (Olbers, 1976; McComas and Müller, 1981; Henyey et al., 1986), as follows. The second method is based on internal wave energy levels directly, as used in the internal wave model IDEMIX (Olbers and Eden, 2013). Internal wave energy levels





are calculated from velocity time series. To quantify the local/near-field effect of waves on turbulence, we consider only the internal wave continuum and semidiurnal baroclinic tides of higher vertical modes. The third method applies the strain-based finestructure parameterization (Gregg, 1989; Wijesekera et al., 1993; Polzin et al., 2014, and references therein) to vertical profiles of temperature and salinity. All dissipation rate results will be compared horizontally and vertically to obtain the contribution of internal wave breaking to the overall dissipation rates.

## 3.1 Total dissipation rate estimates from Thorpe Scales

Total dissipation rates of turbulent kinetic energy are inferred from potential density profiles by analysing Thorpe scales (Thorpe, 1977), meaning the mean sizes of the energy-containing overturnings (Fernández Castro et al., 2022). This vertical scale is defined inside an unstable segment as the root-mean-square of the required vertical displacement of water parcels to form stable stratification. The Thorpe length scale $L_{\mathrm{T}}$ is linearly related to the Ozmidov scale $L_{\mathrm{O}}$, at which buoyancy becomes
important for eddies. If both scales reach similar lengths, the overturns efficiently interact with buoyancy forces and transport mass against the stratification, i.e. pushing lighter water down or bringing denser water up (Fernández Castro et al., 2022). The Ozmidov scale is calculated as $L_{\mathrm{O}} = \varepsilon^{1/2} N^{-3/2}$ (Dillon, 1982), dependent on dissipation rate $\varepsilon$ in units of $\mathrm{W\,kg}^{-1}$, the conversion of turbulent kinetic energy to heat, and the buoyancy frequency $N$ in units of $\mathrm{rad\,s}^{-1}$, describing the vertical stratification.

The overturns, deviations from a stable water column, can be the result of any turbulent event, making this approach blind to the exact process leading to turbulence. Therefore, the dissipation rate $\varepsilon$ in the Ozmidov scale definition equals the total dissipation rate $\varepsilon_{\mathrm{total}}$. This distinction is important, as other methods for quantifying turbulence applied later in this study are not process-independent. The linear relation between Ozmidov and Thorpe scale is defined empirically and slightly varies between studies (Dillon, 1982; Ferron et al., 1998; Voet et al., 2015), but remains close to $1$. Because we lack observations
to compare results of the Thorpe scale approach to direct turbulence measurements, we refer here to the literature value of $L_{\mathrm{O}} = 0.8 L_{\mathrm{T}}$, which is also used in Thorpe scale analysis of a dense water overflow in Storfjorden, located at high latitudes (Fer et al., 2004). This value is comparable to the choice of $0.76$ by (North et al., 2018) for their study of the Denmark Strait overflow. Density noise is estimated to be $3 \times 10^{-4}\,\mathrm{kg\,m}^{-3}$, meaning overturns with smaller top-to-bottom density differences are rejected, in order to exclude spurious overturns due to measurement uncertainty in the profiles. This combines to the relation


$$\varepsilon_{\mathrm{total,\,Thorpe}} = 0.8^2 L_{\mathrm{T}}^2 N^3. \tag{1}$$

For each overturn, partwise constant buoyancy frequency $N$ is calculated using the Gibbs Seawater (GSW) Oceanographic Toolbox (McDougall and Barker, 2011; Firing et al., 2021), a thermodynamically consistent formulation based on the Gibbs function. Where overturns are not detected and consequently no dissipation rates, we assume for averaging purposes a back-
ground dissipation rate of $10^{-10}\,\mathrm{W\,kg}^{-1}$. As turbulence consists of a sequence of low- and high-energetic events, we use an arithmetic average to estimate time-averaged dissipation rates. All profiles of total dissipation rates are averaged arithmetically inside bins of $0.5°$ longitude across the slope. In the vertical, we keep the resolution of the CTD profiles of $2\,\mathrm{m}$.



## 3.2 Wave-induced dissipation rate estimates from squared wave energy

We will calculate dissipation rates, induced by internal gravity waves, $\varepsilon_{\mathrm{IGW}}$ from internal wave energy levels. Internal wave
energy levels themselves are calculated from moored horizontal velocity time series $u$ and $v$, based on spectral methods. The vertical velocity is assumed to be small against the horizontal components and is neglected. The complex horizontal velocity $u + iv$ is viewed as the sum of clockwise and counterclockwise rotating components. Rotary spectra are calculated from complex velocity time series using the multitaper method (Thomson, 1982; Cokelaer and Hasch, 2017). This method repeats spectral calculations of the complex time series in tapered sliding windows. The window width is determined by the
product-bandwidth $P$, which effectively means frequencies inside a window of $2P - 1$ Fourier coefficients are smoothed. We chose a value of product-bandwidth $P = 10$ to balance frequency resolution and noise reduction. Accuracy of the multitaper method is checked by integrating over the full spectrum and comparing the result to velocity time series variance. Velocity spectra are divided by 2 to yield horizontal kinetic energy densities. Both rotary components are added to form spectra $\mathcal{U}(z,\omega)$ of full horizontal kinetic energy (see the example in Fig. 2a). Velocity measurements every 2 hours correspond to a maximum
frequency resolution of 6 cycles per day (abbreviated as cpd). The resulting spectra have the general shape of a plateau at low frequencies and an exponential decay towards high frequencies. On top are peaks, most pronounced at diurnal (1 cpd), and semidiurnal (2 cpd) frequencies. This is shown exemplarily in Fig. 2b for a spectrum derived from velocity time series measured at mooring **B** (63.51° S, 51.64° W) at a depth of 1513 m.

### 3.2.1 Wave energy available for local dissipation

However, not all of the observed horizontal kinetic energy can be attributed to internal gravity waves. Additionally, waves contribute their energy only in part to near-field mixing. Therefore, it is necessary to clearly distinguish the underlying energy sources and transfer processes that together produce the observed horizontal kinetic energy spectra (Fig. 2a). The generation of internal waves happens at large scales, for example from interaction of (tidal) currents with the rough sea floor. Wave–wave interactions (for example parametric subharmonic instabilities (Olbers et al., 2020), wave–topography interactions, or
wave–mean-flow interactions transfer (Musgrave et al., 2022) the energy to ever smaller scales, where the likelihood for wave breaking increases. We identify the smooth exponential decay of the background inside the frequency range Coriolis frequency $f$ to buoyancy frequency $N$ with the so-called internal wave continuum (Munk, 1981). An attempt of finding a general model for this spectrum is done in the Garrett-Munk model of the internal wave energy spectrum (Garrett and Munk, 1972, 1975). The sharp peaks are the result of overlapping depth-independent barotropic and depth-varying baroclinic tides at their respective
frequencies.

The superposition of propagating internal waves, reflecting repeatedly at the surface and at the bottom, can be viewed as standing vertical waves or modes. The higher the mode number, the smaller the scales, implying that the time scales of wave–wave interactions, which randomize the wave phase, and the travel times between ocean surface and bottom become comparable. In other words, before a standing wave can even form, the nonlinear processes have made the wave incoherent
(Olbers, 1983). This description is especially useful for representing weakly dissipative waves of low mode numbers, which





can transport energy over long distances (Rainville and Pinkel, 2006). Highly dissipative waves of high modes are less well represented, as their travelled distance may be shorter than the distance to the next reflecting plane, e.g. the sea floor or the surface. A summary of the current state of knowledge about mixing by topographically-generated internal waves is given in Musgrave et al. (2022).

Most of the energy at tidal frequencies is contained in the barotropic tide and baroclinic tides of low modes (Falahat et al., 2014). The barotropic tide dissipates energy due to bottom drag (Egbert and Ray, 2003), creating turbulence in a bottom boundary layer. This process overlaps with bottom friction of the gravity current mean flow, creating a homogenously mixed bottom boundary layer. Therefore, higher energy dissipation directly above the sea floor does not lead to changes in stratification, as the bottom boundary layer is already completely mixed (see results in Sect. 4). The energy dissipation of the barotropic tide is 170 thus neglected here. The energy of baroclinic tides is partially transferred to the continuum by the interaction with other waves, topography, or the mean flow, leading to an increase of energy density in the wave continuum.

Of the observed energy at tidal frequencies, mostly the energy contained in higher modes contributes to local turbulence (Falahat et al., 2014), as they are more likely to break.

To accurately estimate wave energy available for local dissipation, we first consider energy in the internal wave continuum 175 and then in the baroclinic tides of higher modes. To split the spectra into continuum and tidal peaks, we calculate the energy of the semidiurnal tidal peaks and subtract it from the kinetic energy density spectrum. For every frequency of the most energetic semidiurnal tidal frequencies, M2, S2, N2, K2 (Padman et al., 2002), we define a peak width $[\omega_{i-P}, \omega_{i+P}]$, dependent on the multitaper product bandwidth $P$. Overlapping frequency ranges around close tidal frequencies are combined. Values of the internal wave continuum spectra at tidal frequencies are defined as the minimum of the peak interval edges

$$\min\Big(\mathcal{U}(\omega_{i-P}), \mathcal{U}(\omega_{i+P})\Big). \tag{2}$$

From this, the wave energy at semidiurnal tidal frequencies exceeding the continuous background can be computed. Subtracting it from the integrated horizontal kinetic energy $\int_f^N \mathcal{U}(z, \omega) \, d\omega$ yields the energy of the internal wave continuum.

### 3.2.2 Conversion from horizontal kinetic energy spectrum to total energy spectrum

For estimating total wave energy, we have to, additionally to the horizontal kinetic energy, consider the wave-induced available 185 potential energy associated with raised isopycnals. Because of low vertical resolution of moored hydrographic measurements, we cannot quantify isopycnal displacement directly. The mooring data provides horizontal kinetic energy spectra $\mathcal{U}$. We exploit here the dispersion relation and the eigenvector (polarization vector) notation for a superposition of linear, random internal waves to derive the required total energy spectra $\mathcal{E}$ (Olbers et al., 2012, Chap. 7.2.2; Pollmann, 2017, Chap. 5.2). Further explanations can be found in App. A. The resulting relation as function of frequency and depth is

$$\mathcal{E}(z, \omega) = 2 \frac{N(z)^2 - f^2}{N(z)^2 - \omega^2} \frac{\omega^2}{\omega^2 + f^2} \mathcal{U}(z, \omega). \tag{3}$$

Before we are able to use Eq. (3) to convert our measured horizontal energy spectra to total energy spectra, we have to determine appropriate values for the buoyancy frequency $N(z)$ at the measurement location and depth of each spectrum. Because $\mathcal{U}(z, \omega)$





represents a time-averaged wave-energy spectrum across the measurement time period, $N(z)$ in equation Eq. (3) must also represent a time-average. Therefore, for each mooring, we select all CTD profiles within a $20\,\mathrm{km}$ radius at each mooring. This

results in 9 to 27 $N^2$ profiles at each mooring site. To compensate slightly different depths at the profile locations, for every mooring location all corresponding profiles are aligned by converting them to distance from the sea floor. Any irregularities close to the sea surface can be ignored, as we are only interested in the processes close to the sea floor. All $N(z)^2$ profiles are smoothed by convolution with a 32-point wide Hanning window, averaged at each mooring, and taken the root of to yield average $N(z)$ profiles. This is done to average over small unstable stratified regions, in which $N(z)$ would be imaginary.

Inserting $N$ and $f$ in Eq. (3) allows now the calculation of $\mathcal{E}(z,\omega)$ from $\mathcal{U}(z,\omega)$, which are both shown exemplarily in Fig. 2b. Because of the measurement period of 2 hours, we do not resolve high-frequency waves faster than $6\,\mathrm{cpd}$. But internal waves are expected up to a frequency of $N$, which in our case always exceeds the resolved frequencies. Buoyancy frequency varies between average values at the rotor current meter locations of $8.4\,\mathrm{cpd} \approx 0.6 \times 10^{-3}\,\mathrm{rad\,s^{-1}}$ to $28.2\,\mathrm{cpd} \approx 2.1 \times 10^{-3}\,\mathrm{rad\,s^{-1}}$.

To include the energy contribution of internal waves faster than $6\,\mathrm{cpd}$, all total kinetic energy spectra $\mathcal{E}$ are extended up

to $N$ with constant spectral slope (see Fig. 2b as an example). The slope is determined by fitting a power law to the tail of the unaltered horizontal kinetic energy spectrum $\mathcal{U}$. This is done to minimize potential errors introduced by the energy conversion factor in Eq. (3), as according to theory the spectral slopes of each energy type are identical. The resulting slopes average to $-1.7 \pm 0.45$ and are therefore on average slightly lower than the theoretical spectral slope of $-2$ in the Garrett-Munk spectrum. The fitted spectral slopes show no discernible dependency on local buoyancy frequency or water depth. And while

we hypothesized that the deviation from the canonical slope correlates with instrument height above the sea floor, we did not observe any relation in the long-term averages (not shown).

A second fit determines the optimal position of the extension of $\mathcal{E}$ on the $y$-axis. The full energy level of the internal wave continuum is calculated by

$$E(z) = \int\limits_{f}^{N} \mathcal{E}(z,\omega)\,\mathrm{d}\omega. \tag{4}$$

Integrating $\mathcal{E}$ over a smaller frequency band yields the energy contained in waves at frequencies inside the band. The spectral extension up to the local buoyancy frequency leads to an energy increase between $5.9\%$ at mooring **A**, $29\,\mathrm{m}$ above the sea floor or in $614\,\mathrm{m}$ depth, to $38.4\%$ at mooring **E**, $91\,\mathrm{m}$ above the sea floor or in $3299\,\mathrm{m}$ depth, compared to using the instrument resolution of $6\,\mathrm{cpd}$ as the integration boundary. Over all mooring measurements, the spectral extension is responsible for an energy increase of $20.13\% \pm 7.87\%$.

**3.2.3 Energy contributions of baroclinic tides**

After including the internal wave energy contribution of the continuum, we turn to the energy contribution of semidiurnal internal tides. We calculate baroclinic tidal energies by first estimating the energy of the barotropic tides and subtracting that from the observed total tidal energy. We estimate the barotropic tide by combining results of the Circum-Antarctic Tidal Simulation (CATS) model (Padman et al., 2002; Howard et al., 2019) with a measurement-based approach, relying on depth-





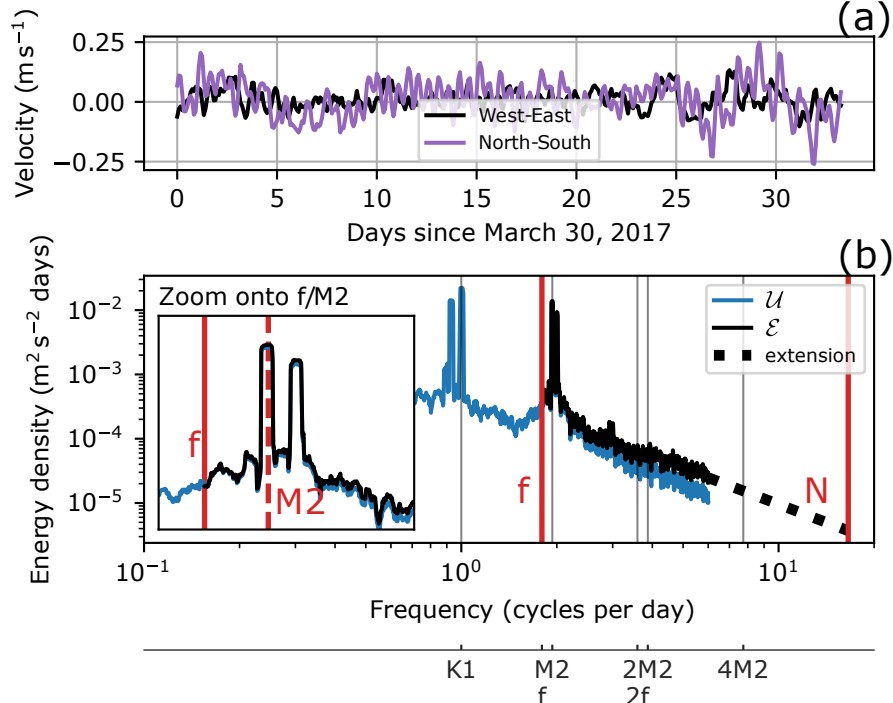

**Figure 2. (a)** Excerpt of a velocity time series, measured at mooring **B** ($63.51°$ S, $51.64°$ W) at a depth of $1513\,\mathrm{m}$, $143\,\mathrm{m}$ above ground. **(b)** Corresponding kinetic horizontal energy spectrum $\mathcal{U}$ in blue. The black line shows the derived total energy spectrum $\mathcal{E}$ in the linear wave range, Coriolis frequency $f$ and buoyancy frequency $N$ are both marked by vertical red lines. The black dotted line shows the spectral extension up to $N$. In the example, the slope is of constant value $s = -1.9$. The inset figure shows a zoom on the semidiurnal tidal frequencies, with the most prominent frequency M2 drawn as a red dashed line.

variations in the baroclinic tide. Because we observe at some mooring locations lower total tidal energy than what the barotropic tidal CATS model predicts, we assume no energy in baroclinic modes at one instrument depth at 4 locations: at mooring **A**, **B**, **E** and **G**. See App. B for further details.

As stated before in Sect. 3.2.1, mostly energy contained in higher vertical modes sources near-field turbulence (Falahat et al., 2014). Therefore, we have to split the baroclinic energy further into its distribution over the modes. Without the necessary
instrument density to resolve vertical modes ourselves, we refer here to results of previous studies. St. Laurent et al. (2002) use a parameterization for internal wave energy flux in a tidal model to estimate a global average for the local dissipation efficiency of baroclinic tides $q \approx 0.3$. Vic et al. (2019) compute ratios of energy in the fourth and higher M2 modes to the total M2 energy, $q = E_{\mathrm{M2}}^{4-\infty}/E_{\mathrm{M2}}^{1-\infty}$. Based on a global model of the M2 internal tide, combined with satellite and in situ measurements, they find for the Weddell Sea continental slope a wide range of ratios, ranging from 0 to 0.7. Without any clear pattern in
their results, we assume for our analysis the global average ratio of 0.3, which is still in agreement with the local dissipation efficiency estimations of Vic et al. (2019).





We use the local dissipation efficiency result to scale the baroclinic energy at every semidiurnal tidal frequency accordingly. The calculated baroclinic energy in higher vertical modes is added to the previously derived energy in the internal wave continuum to yield for each measurement location the full wave energy available for local dissipation. Over all moored mea-
surements, the semidiurnal baroclinic tide increases the full wave energy about $10\%$, with a standard deviation of the same order of magnitude. The highest energy increase with $30.8\%$ is measured at mooring **C**, at a depth of $2343\,\mathrm{m}$ or $150\,\mathrm{m}$ above the sea floor.

### 3.2.4 Dissipation rate from internal wave energy levels

The parameterized dissipation of internal wave energy is a function of the total energy squared (Olbers, 1976; McComas and
Müller, 1981; Henyey et al., 1986). This parameterization is based on wave–wave interaction theory and scaling laws, and assumes that non-linear interactions between waves always transport energy towards higher wave numbers at a rate independent of the wave number itself. Therefore, to calculate how much energy is transformed from internal waves into turbulence, it is possible to look at more easily observable lower wave numbers. The underlying assumptions are validated in numerical evaluations of the scattering integral for wave–wave interactions (Eden et al., 2019; Dematteis and Lvov, 2021). We adapt the
formulation used in the internal gravity wave model IDEMIX (Olbers and Eden, 2013, Eq. 18) and combine the previously derived internal wave energy levels in the $f$–$N$ frequency range with stratification to calculate wave-induced turbulent dissipation rates:

$$\varepsilon_{\text{IGW, IDEMIX}} = \frac{1}{1+\Gamma}\mu_0 f_{\mathrm{e}}\frac{m_\star^2 E^2}{N^2} \tag{5}$$

with the constant mixing coefficient $\Gamma = 0.2$. Although this value and its variability is widely discussed (Gregg et al., 2018),
we use for simplicity the original value of Osborn (1980). The effective Coriolis frequency $f_{\mathrm{e}}$ is defined as

$$f_{\mathrm{e}} = |f|\operatorname{arccosh}\frac{N}{|f|}. \tag{6}$$

As the mooring array only covers less than $1°$ in latitude, we use here a constant Coriolis frequency of $|f| \approx 1.3 \times 10^{-4}\,\mathrm{rad\,s^{-1}}$, which corresponds to around $1.8\,\mathrm{cpd}$. The parameter $\mu_0$ is related to the dissipation of wave energy associated with spectral energy fluxes by wave–wave interactions and $m_\star$ is the wavenumber scale or roll-off wavenumber, which together with the
spectral slope determines the shape of the vertical wavenumber energy spectrum (Pollmann, 2020). Although $m_\star$ is not generally constant in time and space, Pollmann (2020, Fig. 4) observe in the Southern Ocean only small deviations from the canonical $m_\star = 0.01\,\mathrm{rad\,m^{-1}}$ in the Garrett-Munk model (Garrett and Munk, 1972, 1975). We ignore any seasonal variability in $m_\star$, as we only consider long-time averages. For the empirical parameter $\mu_0$, Pollmann et al. (2017) find the best alignment between model outcomes and Argo-float-based estimates of internal wave energy and its dissipation for a value of $\mu_0 = 1/3$,
which we consequently use for this analysis. For the required information about the local buoyancy frequency, we consider the previously calculated $N(z)^2$ values (see Sect. 3.2.2).

We estimate the numerical uncertainty of $\varepsilon_{\text{IGW}}$ from the uncertainties in buoyancy frequency $\Delta N$ and energy level $\Delta E$. As dissipation rate measurements usually follow an approximate log-normal distribution (Whalen, 2021), we calculate the error



to the dissipation rate magnitude instead of the value itself. Further details are presented in App. C. We want to note that this approach cannot quantify the additional uncertainty associated with the many assumptions needed in this method. A discussion of the method uncertainties is presented in Sect. 5.1.

To our knowledge, we are the first one to apply this method to estimate wave-induced dissipation rates from velocity time series. Le Boyer and Alford (2021) make similar approximations and estimate $\varepsilon$ from velocity spectra as well, but use proportional scaling of the Garrett-Munk model, instead of direct estimations from Eq. (5).

## 3.3 Wave-induced dissipation rate estimates from finestructure parameterization

The second method to estimate the wave-induced turbulence $\varepsilon_{\text{IGW}}$ is called finestructure or finescale parameterization (Gregg, 1989; Polzin et al., 2014) and is calculated from vertical hydrographic profiles and, where available, the corresponding velocity profiles. It parameterizes the dissipation rate in dependence of shear, the vertical gradient of horizontal velocity, and/or strain, the vertical gradient of vertical isopycnal displacement. This method is based on the Garrett-Munk model with similar assumptions as the previous method: the variance at small vertical wave numbers can be used to infer the energy transport at very large wave numbers to turbulent scales. The detailed theoretical background of the finestructure parameterization can for example be found in Kunze et al. (2006). We will present here only the necessary numerical steps to get wave-induced dissipation rate $\varepsilon_{\text{IGW}}$ estimations.

Profiles are divided into half-overlapping $250\,\text{m}$ segments with a $125\,\text{m}$ spacing. This results in integration boundaries for shear from $83\,\text{m}$ to $250\,\text{m}$ scales and for strain from $83\,\text{m}$ to $12\,\text{m}$. The centre of the lowest segment is chosen to be half the spacing, $62\,\text{m}$, above the sea floor to balance the size of the lowest averaging window with the lowest data point altitude above ground. If velocity or shear measurements are not available, wave-induced dissipation rate $\varepsilon_{\text{IGW}}$ for each segment can be estimated from vertical gradients of strain $\zeta_z$ (Wijesekera et al., 1993)

$$\zeta_z = \frac{N^2(z) - N^2_{\text{bg}}(z)}{\overline{N^2}}. \tag{7}$$

$N^2(z)$ is the measured buoyancy frequency, while $N^2_{\text{bg}}(z)$ is the smooth background stratification calculated by the adiabatic levelling method, originally by Bray and Fofonoff (1981) and recommended using in Polzin et al. (2014). $\overline{N^2}$ is the segment-averaged squared buoyancy frequency. The dissipation rate itself, using the notation from Whalen et al. (2015), is then

$$\varepsilon_{\text{IGW, fine}} = \varepsilon_0 \frac{\overline{N^2}}{N_0^2} \frac{\langle \zeta_z^2 \rangle^2}{\langle \zeta_{z\text{GM}}^2 \rangle^2} L(f, N)\, h(R_\omega) \tag{8}$$

with $\varepsilon_0 = 6.73 \times 10^{-10}\,\text{W kg}^{-1}$ and $N_0 = 5.2 \times 10^{-3}\,\text{rad s}^{-1}$ being reference values of the Garrett-Munk model for internal waves (Munk, 1981, Sect. 9.9.1). $\langle \zeta_z^2 \rangle$ is observed strain variance, $\langle \zeta_{z\text{GM}}^2 \rangle$ is the Garrett-Munk model strain variance. Because the Garrett-Munk model was originally developed for $30°\,\text{N}$, we have to use a correction factor to adapt the method to the latitudes of our data around $64°\,\text{S}$:

$$L(f, N) = \frac{f \operatorname{arccosh}\left(\frac{\overline{N}}{f}\right)}{f_{30°} \operatorname{arccosh}\left(\frac{N_0}{f_{30°}}\right)}. \tag{9}$$



The second correctional factor

$$h\left(R_\omega\right) = \frac{R_\omega\left(R_\omega + 1\right)}{6\sqrt{2}\sqrt{R_\omega - 1}} \tag{10}$$

depends on the shear-to-strain variance ratio $R_\omega$

$$R_\omega = \frac{\langle U_z^2\rangle}{N^2\langle\zeta_z^2\rangle}. \tag{11}$$

with the observed shear variance $\langle U_z^2\rangle$, averaged over the resolved wave numbers. For a single wave, this is equivalent to the ratio of horizontal turbulent kinetic to available potential energy $R_\omega = \frac{\text{HKE}}{\text{APE}}$ (Kunze et al., 2006). Without shear data, $R_\omega$ cannot be computed and has to be assumed. The Garrett-Munk model value prescribes $R_\omega = 3$, with $h(R_\omega = 3) = 1$. But global observational data suggest an average ratio closer to $R_\omega = 7$ (Kunze et al., 2006), with $h(R_\omega = 7) \approx 2.694$. From the single cruise PS129, where both hydrographic and velocity profiles are available from CTD and LADCP measurements, we can compute $R_\omega$ in the northwestern Weddell Sea. This yields an approximately log-normal $R_\omega$ distribution with an arithmetic mean and standard deviation of $7.9 \pm 10.3$, and supports our choice to use $R_\omega = 7$ in the strain-dependent formulation. But where both hydrographic and velocity profiles are available, we are able to compare the results of Eq. (8) with well-chosen corrective terms to the results of the formulation directly dependent on strain and shear (see App. D). In this limited data set, their ratio is close to 1 for many segments, which supports the use of Eq. (8) for estimating dissipation rates from all CTD profiles along the transect.

Because the finestructure parameterization is applied on vertical segments, this method can only consider vertical modes of internal tides with wave numbers smaller than the segment length of $250\,\text{m}$. Luckily, these observed higher modes contain the energy that is dissipated locally through turbulence. $\varepsilon_{\text{IGW, fine}}$ in Eq. (8) describes therefore the combined effect of the internal wave continuum and internal tides, the same as $\varepsilon_{\text{IGW, IDEMIX}}$ in Eq. (5). Finestructure parameterization is implemented with the *mixsea* package for python (Voet et al., 2023). More details about the method can be found in the *mixsea* package documentation. To allow for vertical and horizontal averaging, we assume the same background dissipation rate as before of $10^{-10}\,\text{W}\,\text{kg}^{-1}$. All profiles of wave-induced dissipation rates $\varepsilon_{\text{IGW, fine}}$ are averaged arithmetically inside bins of $0.5°$ longitude across the slope. A discussion of the method uncertainties is presented in Sect. 5.1.

## 4 Results

The extent of the Weddell Sea Bottom Water gravity current is defined as the height of the neutral density surface $\gamma^n = 28.40\,\text{kg}\,\text{m}^{-3}$ (Naveira Garabato et al., 2002b). Gravity current thickness varies up to $100\,\text{m}$ between expeditions, although all CTD measurements were collected in the same season of austral summer. The gravity current flows across the Joinville transect on average in north-westerly direction, consistent with the direction of the isobaths. All except the deepest moorings **F** and **G** show aligned mean current directions along the slope (not shown), with slightly higher current speeds towards the sea floor (see Fig. 3). The strongest mean velocities are measured by the bottommost current meter at mooring **B** ($51.6°$ W) at a water depth of $1656\,\text{m}$ and by the bottommost current meter at mooring **D** ($50.09°$ W) at a water depth of $2757\,\text{m}$. This is



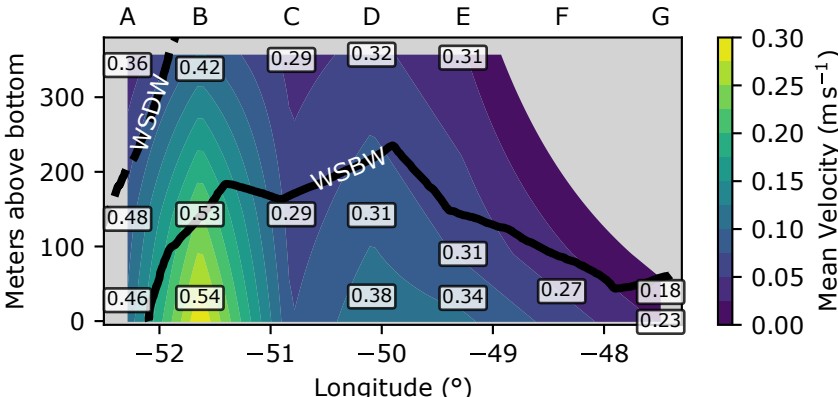

**Figure 3.** Flow field across the continental slope. Mean flow is linearly interpolated between measurement locations and displayed as amplitude in color. Grey boxes show rotor current meter positions, labelled with the peak current speed during the measurement period in units of $\mathrm{m\,s^{-1}}$. Moorings are labelled **A–G**. Black lines denote water mass boundaries, derived from neutral density. The definition of Weddell Sea Bottom Water (WSBW) is also taken as the extent of the gravity current.

interpreted as at least one core of the gravity current, which shows mean flow velocities around $0.30\,\mathrm{m\,s^{-1}}$ and reaches peak velocities of $0.54\,\mathrm{m\,s^{-1}}$ (Fig. 3). Based on the CTD profiles taken between 1989 and 1998, listed in Table 1, a flow field with two cores of the Weddell Sea Bottom Water gravity current is already identified by Fahrbach et al. (2001). A detailed analysis of the time-varying flow field and height of the gravity current can be found in Llanillo et al. (2023). But even without a further analysis, we see from Fig. 3 a decrease in current speed starting from the upper gravity core towards the deep sea. The stark differences between low mean velocities, but high peak velocities at mooring **A** can be explained by a weak current, but strong tides. In the deep sea at mooring **G**, mean as well as peak flow is small due to weak tides and the location at the outermost edge of the gravity current.

The stratification of the lowermost $400\,\mathrm{m}$ varies along the transect. Shallower waters towards the shelf show more variability, with buoyancy frequencies fluctuating around $1.1 \times 10^{-3}\,\mathrm{rad\,s^{-1}}$. Going down the continental slope, stratification between $400\,\mathrm{m}$ to $200\,\mathrm{m}$ above the sea floor, above the gravity current, decreases to be almost constant at $0.3 \times 10^{-3}\,\mathrm{rad\,s^{-1}}$. Inside the gravity current, stratification increases up to a maximum before buoyancy frequency drops to almost zero directly above the sea floor, indicating a homogenously mixed bottom boundary layer.

Thorpe scale analysis reveals the across-slope pattern of turbulence (Fig. 4). Across the slope we see a quiescent region of the water column, above the gravity current, with dissipation rates $\varepsilon_{\mathrm{total}}$ of $10^{-9}\,\mathrm{W\,kg^{-1}}$ down to the background threshold of $10^{-10}\,\mathrm{W\,kg^{-1}}$. The segment of high turbulence at $47.6^{\circ}\,\mathrm{W}$, exceeding $10^{-6}\,\mathrm{W\,kg^{-1}}$, can be traced back to a single outlier profile. Neither a faulty measurement nor a highly turbulent event can be ruled out. These quiescent areas extend into the Weddell Sea Bottom Water gravity current, in which the high vertical resolution of the Thorpe scale analysis allows us to identify two discernible vertical regions inside: a bottom boundary layer (BL) and an interfacial layer (IL) above it. At the



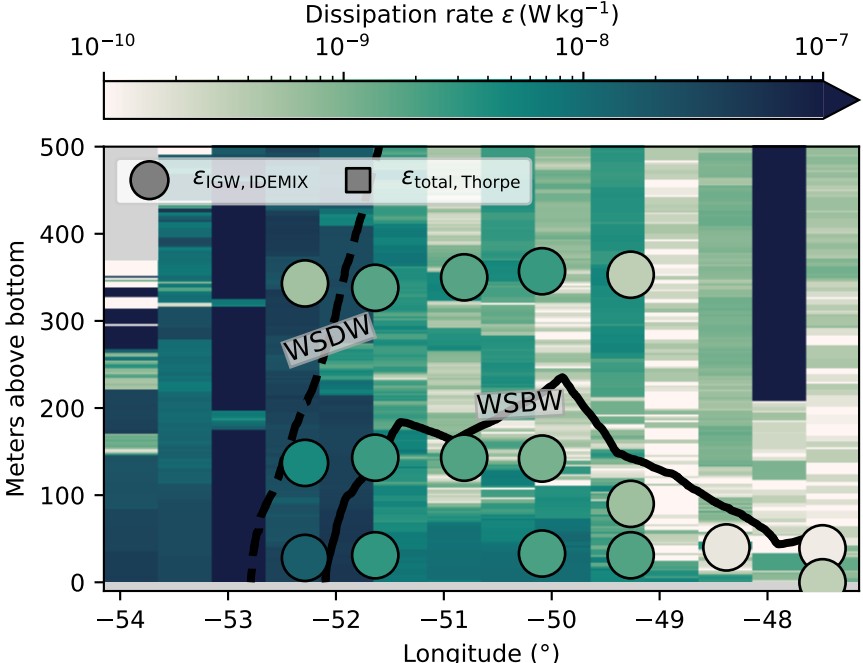

**Figure 4.** Dissipation rates across the continental slope. The background shows total dissipation rates $\varepsilon_{\text{total, Thorpe}}$ from Thorpe scale analysis. Circles show the wave-induced diffusivities $\varepsilon_{\text{IGW, IDEMIX}}$, calculated with Eq. (5) from velocity time series. Black lines denote water mass boundaries, derived from neutral density. The definition of Weddell Sea Bottom Water (WSBW) is also taken as the extent of the gravity current.

interface of the IL layer with the Weddell Sea Deep Water, fewer and smaller overturns than in the BL, are detected, which

results in an $\varepsilon_{\text{total}}$ estimation of around $10^{-9}\,\text{W}\,\text{kg}^{-1}$, not noticeably different from the quiescent middle of the water column. In the BL, close to the sea floor, we measure enhanced dissipation rates $\varepsilon_{\text{total}}$ around $10^{-7}\,\text{W}\,\text{kg}^{-1}$. Buoyancy profiles show that this increased turbulence results in a close to homogenously mixed BL (not shown) The interface of BL to the IL is characterized by strong stratification and a sudden decrease of dissipation rate from the BL to the IL. The bottom boundary layer varies between $20\,\text{m}$ and $100\,\text{m}$ in height across the slope, and decreases in size towards the deep sea (Fig. 4). The same

pattern of decrease in height towards the deep sea is also seen in the gravity current. Similar gravity current structures of increased turbulence near the bottom and weak turbulence across an interface has been found in the Baltic Sea (Umlauf et al., 2007), the Faroe Bank Channel (Fer et al., 2010) and Denmark Strait (Paka et al., 2013; North et al., 2018). At the shelf break around $52°\,\text{W}$, turbulence is elevated throughout the water column, coinciding with strong horizontal and vertical density gradients in the velocity flow field (Llanillo et al., 2023). Here, we observed total dissipation rates up to $\mathcal{O}(10^{-7})\text{W}\,\text{kg}^{-1}$.

Therefore, at the westernmost edge of the gravity current, the two-layer description is not applicable anymore as the water column on the shelf becomes more homogenously mixed.

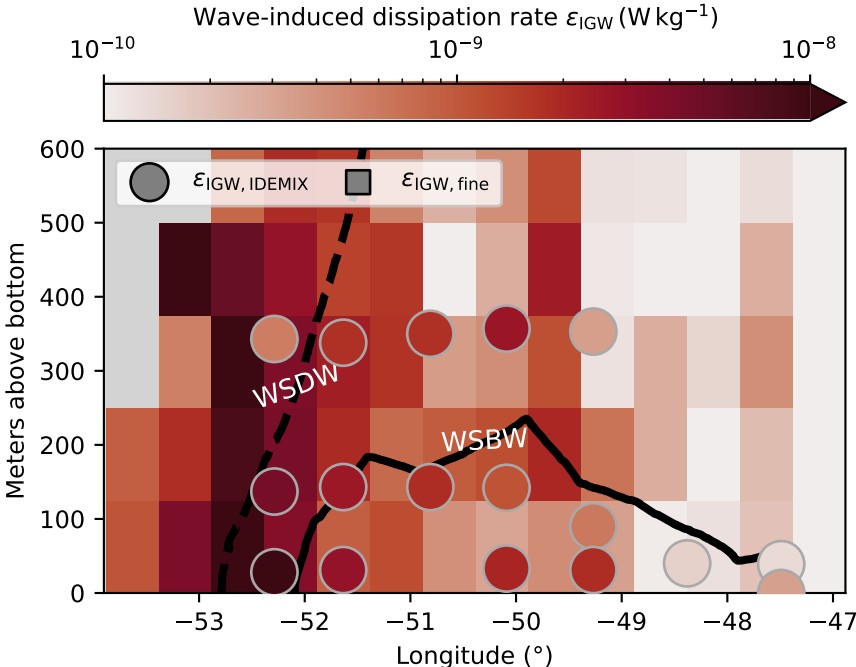

**Figure 5.** Comparison of the two independent methods for estimating internal-wave-induced dissipation rates $\varepsilon_{\text{IGW}}$. Finestructure-based estimations (Sect. 3.3) are shown as rectangles in the background, while wave-energy-based estimations (Sect. 3.2) are shown as circles. Black lines denote water mass boundaries, derived from neutral density. The definition of Weddell Sea Bottom Water (WSBW) is also taken as the extent of the gravity current.

We quantify dissipation rate induced by internal waves by using the two parameterizations described in Sect. 3.2 and Sect. 3.3. The results of Eq. (5), the parameterization based on wave-energy, are shown as circles in Fig. 4 and Fig. 5. The calculated dissipation rates $\varepsilon_{\text{IGW, IDEMIX}}$ range from $10^{-10}\,\text{W}\,\text{kg}^{-1}$ to $10^{-8}\,\text{W}\,\text{kg}^{-1}$ and decrease vertically with distance from

the sea floor. This is observed at all mooring locations and caused by the interaction of low stratification in the BL and vertical changes in wave energy. However, the tidal wave energy features no clear dependence on height above the sea floor. For example, at mooring **A** the most energetic tides are measured closest to the sea floor and decrease in energy with altitude. At mooring **B** this pattern is reversed (see also Fig. B1). Along the transect, we observe a downslope decrease in $\varepsilon_{\text{IGW, IDEMIX}}$ at all instrument levels. This results from weaker internal wave energy further away from the continental shelf. Additionally, we

also see on average per mooring location a downslope decrease in the relative contribution of the semidiurnal baroclinic energy (not shown). This means the energy of the baroclinic tide decreases faster in energy than the internal wave continuum, which leads to it contributing relatively less energy to the overall wave energy available for turbulence.

The results from Eq. (8), the finestructure method, show as well a horizontal decrease in $\varepsilon_{\text{IGW, fine}}$. This is apparent in Fig. 5, with wave energy dissipation rates around $5 \times 10^{-9}\,\text{W}\,\text{kg}^{-1}$ at the shelf break and around $3 \times 10^{-10}\,\text{W}\,\text{kg}^{-1}$ or less in the





deep sea. Compared to the horizontal pattern, vertical changes in $\varepsilon_{\text{IGW, fine}}$ are small. Inside the gravity current, we estimate dissipation rates $\varepsilon_{\text{IGW, fine}}$ to be around $10^{-9}\,\text{W}\,\text{kg}^{-1}$. Measurements outside the gravity current can be both more and less turbulent, with no apparent pattern. The length scale of finestructure method resolution of $125\,\text{m}$ can exceed the two layer structure and small differences cannot be resolved. We want to note that the finestructure parameterization is build on various assumptions, which will be discussed in Sect. 5.1.

Despite describing the same concept of wave-induced dissipation rates $\varepsilon_{\text{IGW}}$, the two presented methods differ in their exact results. This is an expected outcome, because both methods estimate dissipation rate from larger-scale, but different observables. Both methods for wave-induced dissipation rates agree on the horizontal pattern of high wave-induced turbulence towards the shelf and a more quiet water column towards the deep sea. The biggest difference of about 2 orders of magnitude is measured at the westernmost mooring **A**, $320\,\text{m}$ above the sea floor (see Fig. 5). Here, the estimation from long-time velocity

records suggest much less wave-induced turbulence than the finestructure profiles averaged over the corresponding bin. Across all measurements, the dissipation rates calculated from the strain-based finestructure method are generally slightly lower than the values calculated with the IDEMIX parameterization. This becomes especially apparent in the horizontally averaged vertical profile, shown in Fig. 6. For this, we horizontally average dissipation rates arithmetically along levels of altitude over the gravity current core between $48.5°\,\text{W}$ and $51.5°\,\text{W}$. The resulting average for each method is denoted by angular brackets $\langle\,\rangle$.

In the comparison to the results of the Thorpe scale approach, physics demands that wave-induced dissipation rates $\varepsilon_{\text{IGW}}$ are strictly lower than the total dissipation rates $\varepsilon_{\text{total}}$, induced by all possible processes. This is observed over the whole transect, as $\varepsilon_{\text{IGW}}$ is generally lower than the nearest $\varepsilon_{\text{total}}$ measurement and only exceptionally exceeds $\varepsilon_{\text{total}}$ by a margin smaller than the error bounds (for example, the data point at $130\,\text{m}$ in Fig. 6). We see the largest difference between $\varepsilon_{\text{total}}$ and $\varepsilon_{\text{IGW}}$ in the highly turbulent bottom boundary layer (see Fig. 4 and Fig. 6). The extent of the bottom layer is clearly visible from the height

of elevated total dissipation rate $\langle\varepsilon_{\text{total}}\rangle$, around a value of $10^{-8}\,\text{W}\,\text{kg}^{-1}$. In the average profile, the BL reaches a height of $60\,\text{m}$. The large difference there between $\langle\varepsilon_{\text{total}}\rangle$ and $\langle\varepsilon_{\text{IGW}}\rangle$ supports the assumption that the bottom layer is largely mixed by processes outside internal wave breaking, like barotropic tides, convection, or friction of the mean flow with the sea floor. In the presence of a homogenously mixed BL, the concept of wave-induced turbulence loses its meaning, as the BL prevents the propagation of internal waves. The potential shortcomings of the methods will be discussed in the next section.

Higher up, in the intermittent layer, the total and wave-induced dissipation rates become of similar magnitude $\mathcal{O}(10^{-9})$. Outside the gravity current, from about $300\,\text{m}$ above the seafloor and further, we also see an agreement of Thorpe scale and finestructure estimations, congruent with the assumption that the inner water column turbulence is mainly caused by internal waves. The dissipation rates measured above the gravity current from all methods do not change notably from their respective values in the interfacial layer. This fact can be interpreted as such that also in the interfacial layer, internal waves are responsible

for most of the created turbulence. But in this orders-of-magnitude-perspective, we cannot exactly say which percentage of the total turbulence can be ascribed to internal wave turbulence. Nonetheless, we can conclude that tidal and non-tidal internal waves play an important role for vertical mixing inside the interfacial layer.



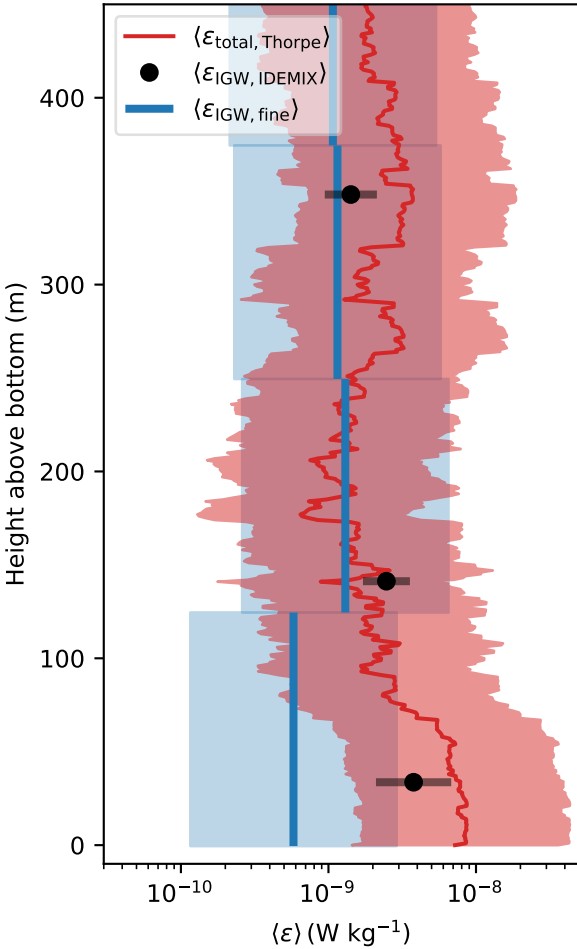

**Figure 6.** Horizontally averaged vertical profiles of total $\langle \varepsilon_{\text{total, Thorpe}} \rangle$ and wave-induced dissipation rates $\langle \varepsilon_{\text{IGW}} \rangle$, distinguished by their method in a second subscript. The Thorpe scale results and their associated uncertainty of a factor of 5 are shown in red. The fixed-depth mooring-based $\varepsilon_{\text{IGW, IDEMIX}}$ are shown as black dots with their numerical error bars. Beware that in comparison to the other uncertainties, these do not stem from the comparison to microstructure results, but are instead calculated with Gaussian error propagation (see App. C). The results of finestructure method $\varepsilon_{\text{IGW, fine}}$ are shown in blue vertical line segments, with their associated uncertainty of 5 as faint blue rectangles.





## 5   Discussion

### 5.1   Uncertainties in the estimations of dissipation rates

For any meaningful comparisons between the computed dissipation rates, we have to include the error margins of each method. Direct microstructure measurements of dissipation rates, which could act as a benchmark, are not available along the Joinville transect. Therefore, we must rely on an understanding of the method scope, numerical error calculations, and published uncertainty estimations of the methods we use. Any comparison we do is necessarily a comparison over spatio-temporal scales, due to the different types of measurements underlying the methods. Whalen (2021) showcase how different scales of averaging can
introduce spurious discrepancies between a factor of 2 up to a factor of 10, depending on the turbulence strength. By using averages over different years and $0.5°$ longitude bins, we try to take into account the inherent large variability of turbulence in time and in space (Gregg et al., 1993; Moum et al., 1995).

For the proportionality constant between Thorpe and Ozmidov scale, we use a tested literature value of $0.8$, but there is evidence that the correlation is not necessarily constant (Mater et al., 2015; Mashayek et al., 2017). Although we follow
standard oceanographic practice, results like Scotti (2015) show that the usual practice may only hold for turbulence from shear-driven flows. They find that, when turbulence is instead driven by the available potential energy of the mean flow (also called convective-driven), the proportionality factor between Thorpe scale and Ozmidov scale is no longer $\mathcal{O}(1)$. Gravity currents are ultimately driven by their available potential energy, but we expect both convection and shear instabilities to occur, due to a non-uniform flow field and breaking internal waves. Without any further knowledge about the underlying processes, the standard
practice is our best estimation of process-independent dissipation rates. During the calculation of $\varepsilon_{\text{total, Thorpe}}$ (see Eq. (1)) small overturns and measurement noise can be undistinguishable and are cut off to not include spurious turbulence, controlled by the density noise parameter. This can lead to a bias against quiescent regions, in which dissipation rate is determined with higher uncertainties. The Thorpe scale method is estimated to be generally within a factor of about 5 to direct microstructure measurements (Dillon, 1982; Ferron et al., 1998; Alford et al., 2006). Although microstructure measurements have their own
associated uncertainties, we take the factor of $5$ here as an uncertainty of the Thorpe scale method itself.

The use of a wave dissipation parameterization from energy in the interfacial layer of gravity currents is generally dismissed in Seim and Fer (2011), due to the prevalence of many turbulent processes not described by the Garrett-Munk model. The finestructure parameterizations assumes that the observed variability stems from internal wave activity alone, which likely breaks down within the gravity current, especially in the bottom layer. The violation of this assumption may mean that the
energy transfer through the frequency spectrum is ill-described. All measured energy spectra resemble the smooth spectral decay associated with an internal wave continuum (see for example the spectrum in Fig. 2b). But from the measured data, we cannot say anything about the spectral shape at higher frequencies, where the effect of non-wave processes like instabilities may be seen. When we apply both methods for wave-induced dissipation rates nonetheless, our results are physically plausible, as estimated $\varepsilon_{\text{IGW}}$ is on average less than estimates of $\varepsilon_{\text{total}}$.

Although the basis of the wave energy method (see Sect. 3.2) is also used in the finestructure parameterization, to the knowledge of the authors, our particular approach of calculating wave-induced dissipation rate from velocity time series was





not yet done by others and comparisons with direct microstructure measurements are not available. From the variability of the $N^2$ profiles and uncertainties in the wave energy calculation, we estimate for the wave energy method numerical uncertainty an average factor of around $1.5$ up to a factor of $2.3$ (see App. C for the calculation). But this approach cannot account for

the errors introduced by the many assumptions necessary for this dissipation rate parameterization from observations of larger scales. This leads to likely erroneously small uncertainties that nevertheless allows us to estimate the additional error we introduce by the method calculations themselves. The numerical uncertainty of $\varepsilon_{\text{IGW, IDEMIX}}$ is generally larger close to the sea floor, as we measured here the highest variability of buoyancy frequency $N$.

For the error of the strain-only parameterization with $R_\omega = 7$ in the Arctic Ocean, Baumann et al. (2023) find $73\%$ of the

estimates are within a factor of $5$ to microstructure observations. This is in agreement with global estimations from Polzin et al. (2014), who see the uncertainty "substantially less" than a factor of $10$, while Whalen et al. (2015) estimate a global agreement between micro- and finestructure mostly between a factor of $2$ to $3$. Together with the following specific biases, we follow the more conservative estimation and use an uncertainty factor of $5$ for the finestructure method. We determine the local ratio of available potential energy and horizontal kinetic energy $R_\omega$ from a subset of the hydrographic profile data, where shear

measurements are available, and take the observed value of $R_\omega = 7.9$, as validation for the literature value of $R_\omega = 7$. With this correction, the two formulations of the finestructure method, dependent on strain only (Eq. (8), and shear and strain (see App. D) differ in almost all segments by a factor less than $10$ (see Fig. D1). For the segment closest to the sea floor, there are two additional error sources. First, due to its vicinity to the sea floor, the lowest segment consists only of $3/4$ of a full averaging window, leading to reduced resolution of small wave numbers. Second, the highly turbulent bottom boundary layer can become

completely mixed. In this environment, where $N^2 \approx 0$, the lack of stratification prevents any propagation of internal waves, as no restoring force can be applied and wave-induced turbulence cannot exist beyond a remnant of a previous turbulent event. Due to its long averaging times and use of average buoyancy frequency profiles, this problem is presumably less prevalent in the $\varepsilon_{\text{IGW, IDEMIX}}$ parameterization based on wave energy, described in Sect. 3.2.

The largest differences between the methods of estimating $\varepsilon_{\text{IGW}}$ is consequently found in the highly turbulent bottom bound-

ary layer (Fig. 5, Fig. 6). It is unclear if these added uncertainties can fully explain the discrepancies, especially in the lowest $125\,\text{m}$ segment above the sea floor. Close to the seafloor, we also observe the largest differences between both wave-induced dissipation rate $\varepsilon_{\text{IGW}}$ estimates and the total dissipation $\varepsilon_{\text{total}}$ (see Fig. 4). Further up the water column, the order of magnitudes of all dissipation estimates become closer and their errors overlap. The similarity of Thorpe scale and finestructure results far above the gravity current points to the general applicability of the methods and the accepted statement that in the inner water

column almost all mixing is due to internal waves.

The large uncertainties in the estimations of dissipation rate also lead us to refrain from calculating turbulent diffusivities. Especially the uncertainty of buoyancy frequency $N$ at the locations of the time series measurements, as well as the extensive discussions surrounding the mixing parameter $\Gamma$ (Gregg et al., 2018, and references therein), would only increase the uncertainty of the results without leading to new insights.





### 5.2 Wave sources inside and outside the $f$–$N$ frequency range

While non-wave processes, like instabilities, are sourced by the local gravitational potential energy of the dense water itself, internal waves can have sources at far distances. Both presented methods to estimate the wave-induced dissipation rates are limited to consider only the effect of linear waves inside the $f$–$N$ frequency range. These internal waves can propagate freely. Both strong diurnal and semidiurnal tides (Foldvik et al., 1990; Robertson, 2001a, b) are present in the Weddell Sea (Fig. 2), of which only semidiurnal tides fall into the $f$–$N$ frequency range. Their interaction with irregular bathymetry create internal tides (Musgrave et al., 2022). Another generation mechanism for waves in the $f$–$N$ frequency range is the interaction of a steady flow with the bathymetry. The necessary condition of steady flow for the generation of internal lee waves (Nikurashin and Ferrari, 2011, 2013) is given by the gravity current. Bathymetry data (Dorschel et al., 2022) shows upstream of the transect multiple smaller ridges, as displayed in Fig. 1b, which could act as wave formation sites.

Because of the limitations of our methods, turbulence induced by waves outside the $f$–$N$ frequency range present a potential error for our results. Poleward of the so-called critical latitude, where $f \geq \omega$, the linear solution to the wave equation becomes exponentially decaying. The resulting waves are termed bottom-trapped internal waves (Falahat and Nycander, 2015). As the linear contribution is so much smaller poleward of the critical latitude, the relative importance of nonlinear terms becomes more important. Nonlinear generation mechanisms include for example unsteady lee waves (Rippeth et al., 2017), and we expect this to happen in the Weddell Sea as well, as the Joinville-transect lies northwards of the semidiurnal critical latitude, but south of the diurnal critical latitude. But the exact contribution of nonlinear wave dissipation is hard to quantify. Although the strong diurnal tides are seen as prominent peaks in the measured spectrum (see Fig. 2b), we hypothesize for our study site that bottom-trapped diurnal tides only enhance turbulence in the already well-mixed bottom layer, without increasing entrainment into the gravity current.

### 5.3 Relation to other studies

When we compare our dissipation rate results to literature values for comparable ocean environments areas, we find that in front of Cape Darnley, another formation site of Antarctic Bottom water, Hirano et al. (2015) use microstructure profilers in a gravity current to calculate dissipation rates $\varepsilon > 10^{-7}\,\mathrm{W\,kg^{1}}$ in a bottom boundary layer of $10\,\mathrm{m}$ vertical scale. As tidal currents at Cape Darnley are limited to small amplitudes, they attribute the main energy source of turbulence to the dense water current itself. For their study in the Faroe bank channel overflow, Seim and Fer (2011) calculate horizontal kinetic energy from velocity spectra and combine that with finestructure analysis. Without calculating wave-induced dissipation in the gravity current, they come to the same conclusion as us, that "internal wave-induced mixing in IL can be significant and should not be ignored". In contrast, North et al. (2018) associate in their study of the Denmark strait overflow high dissipation rates in the IL to shear instabilities, as they observe high shear and Richardson numbers below the critical value of $0.25$. Their observations and conclusions differ from ours, as the Denmark strait overflow displays much higher current velocities up to above $1\,\mathrm{m\,s^{-1}}$ and subsequently shear (North et al., 2018, Fig. 2). In comparison, the Weddell Sea Bottom Water gravity current shows only, even in its cores, mean flow speeds of about $0.3\,\mathrm{m\,s^{-1}}$ and peak velocities of $0.54\,\mathrm{m\,s^{-1}}$ (see Fig. 3 and also Llanillo et al. (2023,



Fig. 7, 8)). Therefore, we expect shear instabilities to play less of a role at our study site. Also for the Denmark Strait overflow, Schaffer et al. (2016) show another example for production of high turbulence, sourced by flow–topography interaction: They observe locally elevated turbulence upstream of small topographic elevations, less than $2\,\text{km}$ wide and $80\,\text{m}$ tall (Schaffer et al., 2016, Fig. 11) and explain it with a mechanism described in Legg (2014): low-mode internal waves interact with isolated topography and propagate upstream, where they get arrested, break and cause turbulence. Due to the strong internal wave field we observe, this process could also happen in the Weddell Sea Bottom Water gravity current. But without further data analysis of along-slope instead of across-slope turbulence patterns, we cannot determine the relevance of this for our study site.

We can compare our results to predicted tidally driven dissipation rates along the transect, for example taken from a global data set of parametrized static tidal mixing (de Lavergne, 2020; de Lavergne et al., 2020). In their study, de Lavergne et al. combine the effects of low modes (mode number 1–10), attenuation by wave–wave interactions, direct breaking of low-mode waves through shoaling, low-mode waves dissipating at critical slopes, scattering of low-mode waves by abyssal hills and generation of high-mode waves by abyssal hills. Our estimation of wave-induced dissipation rates far exceeds in the study region the results of de Lavergne et al. (2020) for purely tidally-induced dissipation rates (not shown). Additionally, de Lavergne et al. predict tidally-induced dissipation rates down to $\varepsilon \in \mathcal{O}(10^{-11})\,\text{W}\,\text{kg}^{-1}$, below the sensitivity threshold of usual turbulence measurements. But a direct comparison between their mixing scheme and our results is difficult, as internal tides lose energy to the continuum through wave–wave interactions and cannot be cleanly isolated in the observations from internal waves of all other frequencies.

Instead of a comparison to a static map of tidally-induced dissipation rate, we might also compare the dynamic output of a climate model coupled with the IDEMIX model (Brüggemann et al., 2024) to directly simulate the propagation of waves and the turbulence they produce. But bottom water production at high latitudes is still highly parametrized in modern climate models to at least mitigate deep water formation biases (Heuzé, 2021). Therefore, the size, stratification and/or physical properties of the simulated Weddell Sea Bottom Water gravity current are either too coarsely resolved or especially on smaller scales too different from reality to allow meaningful comparisons.

### 5.4 Connection to larger scales

We want to set our results in a greater context by comparing them to other dissipation rate measurements along the Weddell Sea Bottom Water gravity current. Further upstream, in the southern Weddell Sea, close to the Filchner-Ronne ice shelf, Fer et al. (2016) observe a bottom layer of $100\,\text{m}$ thickness, in which they measure dissipation rates up to $\varepsilon = 1 \times 10^{-7}\,\text{W}\,\text{kg}^{-1}$ with a microstructure profiler. Because this site lies southwards of the M2 critical latitude, Fer et al. (2016) give the explanation that trapped waves generated on the upper continental slope is the reason for these high diffusivities. Due to the vicinity of the semidiurnal critical latitude, the semidiurnal internal tide cannot propagate far and dissipates its energy in the bottom boundary layer. In comparison, further downslope of the Weddell Sea gravity current, towards the Scotia Sea around the Orkney plateau, Naveira Garabato et al. (2019) observe dissipation rates $10^{-9}$ to $10^{-7}\,\text{W}\,\text{kg}^{-1}$ over the slope in the bottom $250\,\text{m}$, which they attribute to symmetric instabilities. Here, the gravity current core descended already to depths below $3000\,\text{m}$, which



corresponds approximately to measurements east of $50^\circ$ W at our study site on the Joinville transect. At these depths, we see a significant decrease of wave-induced dissipation, congruent with the change to a different dominant mixing process.

The comparison with other dissipation rate estimates in gravity currents suggests that the question of the dominant mixing processes seem to be strongly dependent on location site. For strong wave-induced turbulence, the gravity current must pass a
545 "goldilocks zone", neither too deep nor too shallow with strong tides on a sloping topography. In this environment, the highest dissipation rates are still found in the bottom layer, driven by non-wave processes. But the layer is largely isolated from the ambient water, and mixing therein cannot lead to increased entrainment. Instead, internal waves are responsible for a large fraction of the total dissipation rate at the boundary between ambient water and gravity current interfacial layer.

To embed our results in the larger discussions of a changing climate, we point to results from Strass et al. (2020), who
demonstrate persistent warming of the interior Weddell Sea. Furthermore, they hypothesize that advection-driven temperature rises in Warm Deep Water or Weddell Sea Deep Water could result in enhanced heat transfer into Weddell Sea Bottom Water by entrainment into the gravity current. Zhou et al. (2023) show a 30% volume decrease of Weddell Sea Bottom Water since 1992, most pronounced in the densest water classes. Although the reasons for this are most likely large-scale changes in the Weddell Gyre like multidecadal wind patterns, they can lead to a positive feedback loop: as the density differences between the
gravity current and the surrounding water become smaller, stratification decreases and promotes vertical mixing. This would lead to even more entrainment of lighter water and consequentially accelerated density loss in the Weddell Sea Bottom Water. With the Weddell Sea as an important part of global overturning circulation, changes to Antarctic Bottom Water export could have far-reaching consequences for the stability of the global current system.

## 6 Conclusions

Moored and shipboard observations allow for statistical estimates of dissipation rates in the Weddell Sea Bottom Water gravity current. The goal of this work is to isolate the contribution of internal gravity waves to the total dissipation rates. We present a new application of parameterization of wave-induced dissipation rates from wave energy, derived from time series. The parameterization yields results comparable in value to the long-tested method of finestructure analysis. We observe that the internal-wave-induced dissipation rates are around 2 orders of magnitude stronger in the shallower regions towards the Antarc-
tic continent than in the deep sea. Dissipation rate estimations from Thorpe scales reveal that, although bottom processes causes the highest amount of turbulence in the gravity current, the top of the interfacial layer is at a far enough distance to be largely unaffected by these. We conclude, that in the interfacial layer, internal waves are responsible for a large fraction of the total dissipation rate and therefore for entrainment of ambient waters into the gravity current. With that, our results support the same mechanism for turbulence as concluded in Seim and Fer (2011) for the Faroe Bank Channel overflow. The exact quantification
of the effect of internal tides on turbulence in the gravity current is complicated by large uncertainties, but the general patterns are clear. Our conclusion of wave-induced turbulence as an important contributor to turbulence along the Joinville transect cannot be simply transferred to other gravity currents. Comparison with scientific literature shows that the dominant mixing processes are heavily dependent on the environment. In our case, multiple conditions, like the height and location of the bottom




current not too deep on sloping topography to be still effected by strong tides, come together to facilitate the importance of internal waves.

*Code and data availability.* CTD data is available as refenced in Table 1, mooring data as refenced in Table 2. The code repository for the data analysis and reproduction of figures is published under Pinner (2024). All external software or libraries relevant for the analysis are cited in the corresponding sections.

## Appendix A:  Relation of horizontal kinetic to total wave energy

To calculate the total wave energy just from velocity observations, the contribution of potential energy must be taken from wave theory. For linear internal waves in the frequency range between $f$ and $N$, the relation between the horizontal kinetic energy spectra $\mathcal{U}$ and the total energy spectra $\mathcal{E}$ is known (Olbers, 1983; Olbers et al., 2012, Chap. 7.2.2; Pollmann, 2017, Chap. 5.2, and references therein). Because our velocity measurements contain internal waves of all wave numbers, we denote horizontal kinetic energy spectra $\mathcal{U}(z,\omega)$ only in dependence of wave frequency $\omega$ and depth $z$:

$$\mathcal{U}(z,\omega) = \int 2\frac{N(z)^2-\omega^2}{N(z)^2-f^2}\ \frac{\omega^2+f^2}{\omega^2}\mathcal{E}(z,m,\omega)\mathrm{d}m$$
$$= 2\frac{N(z)^2-\omega^2}{N(z)^2-f^2}\ \frac{\omega^2+f^2}{\omega^2}\mathcal{E}(z,\omega)\int A(m)\mathrm{d}m, \tag{A1}$$

with wave frequency $\omega$, buoyancy frequency $N$, and Coriolis frequency $f$, all in units of $\mathrm{rad\,s^{-1}}$. We assume that we can factor out the wave number $m$ dependency of the spectrum of total energy $\mathcal{E}(z,m,\omega) = \mathcal{E}(z,\omega)A(m)$. The same approach of factorisation is used in the Garrett-Munk model (Munk, 1981), but in contrast, we are not required to make any further assumptions about the form of $\mathcal{E}(z,m,\omega)$ here. As $\int A(m)\mathrm{d}m = 1$ (Pollmann, 2020, Eq. 1–3), we can rearrange to Eq. (3) from the main text

$$\mathcal{E}(z,\omega) = 2\frac{N(z)^2-f^2}{N(z)^2-\omega^2}\ \frac{\omega^2}{\omega^2+f^2}\ \mathcal{U}(z,\omega). \tag{A2}$$

The presented proportional factor between $\mathcal{E}$ and $\mathcal{U}$ diverges in the limit of $\omega \to N$. In comparison, the ratio in the Garrett-Munk model of kinetic to total wave energy approaches 2 in the limit of $\omega \to N$. Kinetic energy and potential energy contribute there equally. In the limit of $\omega \to f$, total wave energy in both frameworks is fully captured by kinetic energy. In our analysis, we decide to use Eq. (A2), but for all resolved frequencies in our measured spectra, the difference to the Garrett-Munk model conversion factor is negligible.

## Appendix B:  Estimating the energy in the baroclinic tide

As one step to calculate from the moored velocity records the total wave energy available for local dissipation, we must separate the energy in the internal baroclinic tidal waves from the depth-independent barotropic tide. To overcome the limitations of the



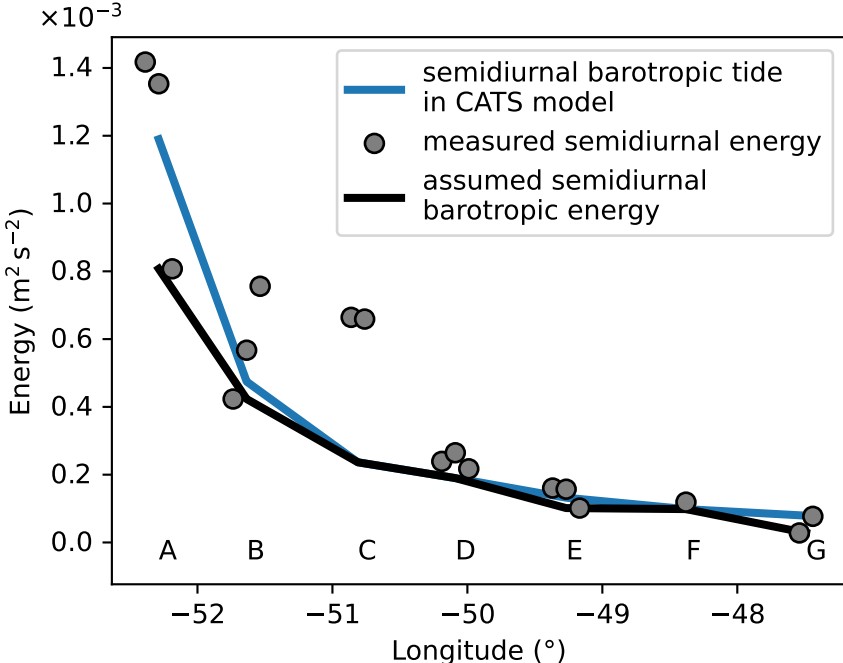

**Figure B1.** Comparison of predicted barotropic kinetic energy by the CATS model, actual measured semidiurnal tidal energies and the following newly assumed best estimate for barotropic horizontal kinetic energy across the continental slope. Measured energies are shifted along the longitude axis for visualization purposes, where for each mooring the measurement closest to the sea floor is to the left. All moorings are labelled **A–G**, according to Table 2.

low vertical resolution of the moorings, we run the Circum-Antarctic Tidal Simulation (CATS) model (Padman et al., 2002; Howard et al., 2019) using Tide Model Driver (Greene et al., 2023) for the duration of the moored velocity measurements. This regional inverse model simulates barotropic tidal horizontal velocities of the strongest constituents in the Southern Ocean (M2, S2, N2, K2, K1, O1, P1, Q1, Mf & Mm) on a 4 km grid. Exactly as with the measured velocities in Sect. 3.2, we calculate
horizontal kinetic energy spectra and integrate over an interval around the semidiurnal frequencies. This predicts barotropic tidal energies for each mooring location. The observed strength of the barotropic tide decreases exponentially from the shelf to the deep sea (Fig. B1). But especially on the shelf, the predicted energy exceeds at some depths the measured tidal energy at semidiurnal frequencies. We believe that this disagreement between model and measurements could not be prevented by the use of a different tidal model. In their comparison of Antarctic ocean tidal models, Sun et al. (2022, Fig. 4) conclude that at
our study site of the Joinville-transect in the northwestern Weddell Sea, in the most dominant M2 constituent any differences of CATS to comparable models are small.

In the case we measure lower tidal kinetic energy (shown as grey dots in Fig. B1) than the predicted barotropic tidal energy in the CATS model (blue line), the model prediction is discarded, and instead we exploit the depth-dependence of the baroclinic





tide. We then take the lowest measured kinetic energy at semidiurnal frequencies at each mooring location as a new best
estimate of the barotropic tide (black line). The result still shows the same exponential decline, strongest on the shelf and
weakest in the deep sea. Subtracting the barotropic energy from the measured tidal energies, yields the baroclinic horizontal
kinetic energies. The dependence of the tidal energy in the vertical axis can be seen in each group of points in Fig. B1, in
which each energy measurement is shifted in longitude according to their distance from the seafloor. We sometimes observed
the highest energy farthest from the bottom (moorings **A**, **C** and **E**), and sometimes closest to the ground (moorings **B** and
**G**). We cannot identify the underlying reason, but hypothesize that this is the effect of vertical and horizontal variations in the
baroclinic tides. Finally, applying the scaling from Eq. (3) converts the baroclinic horizontal kinetic wave energies to baroclinic
total wave energies, which we then use further in Sect. 3.2.3.

## Appendix C: Error of the parameterization from wave energy

We want to repeat that this section only estimates the numerical error we make, and cannot account for the errors introduced by
the many assumptions necessary for this dissipation rate parameterization from observations of larger scales. Repeating Eq. (5)
in the main text

$$\varepsilon_{\text{IGW, IDEMIX}}(E, N) = \frac{1}{1 + \Gamma} \mu_0 |f| \operatorname{arccosh} \frac{N}{|f|} \frac{m_\star^2 E^2}{N^2}, \tag{C1}$$

we want to calculate the uncertainty in our dissipation rate estimates. But from finestructure results in this study and general
literature (Whalen, 2021), we know $\varepsilon$ is approximately log-normal distributed. A symmetric additive error would therefore be
nonsensical. Therefore, we calculate instead the error to the order of magnitude, computed as the common logarithm $\log_{10}$ of
dissipation rate, to achieve a multiplicative error, which is symmetric in $\log$-scale. We constrain ourselves here to only account
for the largest errors introduced by the uncertainties in buoyancy frequency $N$ and wave energy $E$ and neglect uncertainties
of the constants $\Gamma$, $\mu_0$, $f$ and $m_\star$. The method uncertainty $\Delta \log_{10} \varepsilon_{\text{IGW, IDEMIX}}(E, N)$ is then calculated with the ansatz of
Gaussian error propagation, as

$$\left(\Delta \log_{10} \varepsilon_{\text{IGW}}(E, N)\right)^2 = \left(\Delta N \frac{\partial}{\partial N} \log_{10} \varepsilon_{\text{IGW}}(E, N)\right)^2 \qquad + \left(\Delta E \frac{\partial}{\partial E} \log_{10} \varepsilon_{\text{IGW}}(E, N)\right)^2$$

$$= \left(\frac{\Delta N}{\sqrt{N^2 - f^2} \ln(10) \arctan\left(\frac{N}{|f|}\right)} - \frac{2\Delta N}{N \ln(10)}\right)^2 \qquad + \left(\frac{2\Delta E}{E \ln(10)}\right)^2. \tag{C2}$$

All derivations and simplifications are calculated with the symbolic maths library *SymPy* (Meurer et al., 2017).

But determining the uncertainties $\Delta N$ of buoyancy frequency and $\Delta E$ of energy level itself is non-trivial. We take the
uncertainty of $N$ from the variability of $N^2$ in at each mooring location. The standard deviation of the $N^2$ profiles $\Delta N^2$, during
the averaging to estimate a representative stratification (see Sect. 3.2.2), is propagated as a Gaussian error to the corresponding
uncertainty $\Delta N = \frac{1}{2} \left(N^2\right)^{-\frac{1}{2}} \Delta N^2$. The calculation from squared buoyancy frequency $N(z)^2$ is done to allow the averaged
profile to contain small unstable stratified regions, in which $N(z)$ would be imaginary. We motivate the shift from a natural





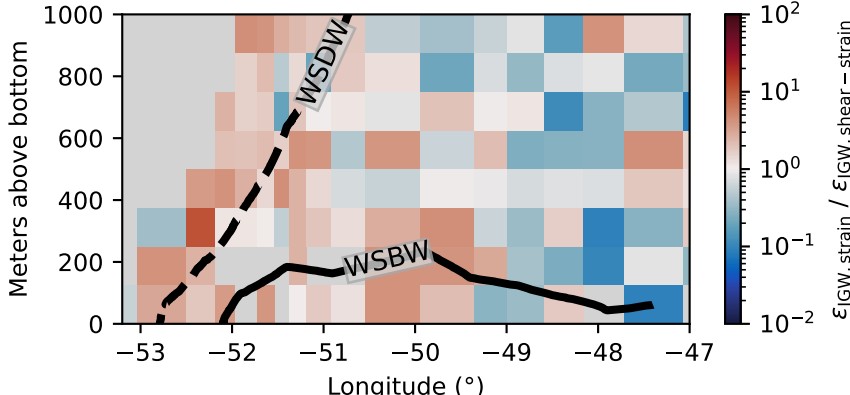

**Figure D1.** Ratio of both finestructure formulations on a logarithmic scale, calculated from measurements taken during the PS129 expedition. The black lines denote the boundaries of water masses, derived from neutral density. The definition of Weddell Sea Bottom Water (WSBW) is also taken as the extent of the gravity current.

variability of $N^2$ to an uncertainty of $N$ to account for the error we make by assuming a constant-in-time buoyancy frequency profile.

Technically, $\Delta E$ is not independent of $\Delta N$, as the buoyancy frequency determines the upper integration boundary of the spectrum in Eq. (4). But Fig. 2b shows that the energy density drops about 2 orders of magnitude towards $N$, while $N$ only varies at maximum one order of magnitude. Therefore, we assume the independence of $\Delta E$ and $\Delta N$ in Eq. (C2). We determine the error of the wave energy calculation $\Delta E$ from the uncertainty of the slope extension, given by the fit algorithm itself, and converting that to a range for the results of every integral over the wave total energy density $\mathcal{E}(\omega)$. This approach may slightly

underestimate the uncertainty $\Delta E$, as the error in the calculation of the rotary spectra (see Sect. 3.2) is neglected. The combined multiplicative uncertainty for the dissipation rate, determined from Eq. (C1) and averaged over all measurement locations, is around 1.5 up to a factor of 2.3. The results of the uncertainty calculations are displayed in Fig. 6 and discussed in Sect. 5.1.

**Appendix D: Shear-strain formulation of finestructure parameterization**

If stratification as well as shear profiles are measured, the finestructure parameterization can be formulated as follows (Kunze

et al., 2006) to calculate wave-induced dissipation rate inside a segment (in the notation of Fine et al. (2021)):

$$\varepsilon_{\text{IGW, fine}} = \varepsilon_0 \frac{\overline{N^2}}{N_0^2} \frac{\langle U_z^2 \rangle^2}{\langle U_{z,\text{GM}}^2 \rangle} L(f,N) h_1(R_\omega). \tag{D1}$$

$\overline{N^2}$ is the segment-averaged squared buoyancy frequency, $\langle U_z^2 \rangle$ is the observed average shear variance over the resolved wave numbers and $\langle U_{z,\text{GM}}^2 \rangle$ is the same expected value from the Garrett-Munk model. The latitudinal correction $L(f,N)$ is the same as in Eq. (9). The correction term $h_1$ differs from the corresponding term Eq. (10) in the formulation in Eq. (8), dependent only



on strain, but is instead as follows:

$$h_1\left(R_\omega\right) = \frac{3\left(R_\omega + 1\right)}{2\sqrt{2}R_\omega\sqrt{R_\omega - 1}}. \tag{D2}$$

The shear-to-strain variance ratio $R_\omega$ can be computed directly for every segment with

$$R_\omega = \frac{\langle U_z^2 \rangle}{N^2 \langle \zeta_z^2 \rangle} \tag{D3}$$

and the segment averaged vertical gradient of strain $\langle \zeta_z^2 \rangle$. The data of the PS129 expedition, where CTD and LADCP profiles were taken, allows us to compare both finestructure formulations. The dissipation rate estimates, from just strain (Eq. (8)) and from shear and strain (Eq. (D1)), will be denoted as $\varepsilon_{\text{IGW, strain}}$ and $\varepsilon_{\text{IGW, shear-strain}}$. Their ratio in the lowermost $1000\,\text{m}$ across the continental slope is displayed in Fig. D1. Inside the gravity current, the strain-based formulation estimates higher dissipation rates than the shear-strain-based formulation. But across the whole transect, no apparent pattern is visible, as the ratio fluctuates seemingly random around 1.

*Author contributions.* **Conceptualization:** Ole Pinner, Torsten Kanzow, Friederike Pollmann; **Formal analysis:** Ole Pinner; **Funding acquisition:** Torsten Kanzow, Friederike Pollmann; **Investigation:** Ole Pinner; **Supervision:** Markus Janout, Torsten Kanzow; **Visualization:** Ole Pinner; **Methodology:** Ole Pinner, Gunnar Voet, Friederike Pollmann; **Writing – original draft:** Ole Pinner; **Writing – review & editing:** Ole Pinner, Friederike Pollmann, Gunnar Voet, Markus Janout, Torsten Kanzow

*Competing interests.* The authors declare that they have no conflict of interest.

*Acknowledgements.* This paper is a contribution to the project T3 *Energy Transfers in Gravity Currents* of the Collaborative Research Centre TRR 181 *Energy Transfers in Atmosphere and Ocean*, funded by the German Research Foundation (DFG) under Grant 27476265. We especially want to thank the Captains and crews of the many voyages of RV *Polarstern* throughout the years on which this publication is based on. Many thanks to Martin Losch for his opinion and expertise given in all T3 & TAC meetings in which this work was discussed. At last, many thanks also to Dirk Olbers, as many equations used here are build on his internal wave analysis and expertise. During the work on this paper, additional useful python packages were *matplotlib* (Hunter, 2007; Caswell et al., 2023), *cartopy* (Met Office, 2010/2015) and *cmocean* (Thyng et al., 2016).



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
