# Peer review of "Internal-wave-induced dissipation rates in the Weddell Sea Bottom Water gravity current"

_EGUsphere, 2024_

## Author Response (AR1)

**Author's response to the interactive discussion**

Ole Pinner (correspondent author)

January 29, 2025

In the following, we will answer each reviewer's comments point-by-point.

**1 st reviewers comments**

Comments can be found at `https://doi.org/10.5194/egusphere-2024-2444-RC1`.
The authors thank the reviewer for the overall very positive assessment of our work as well as for highlighting of inconsistencies, ambiguities and the occasional lack of clarity.

**1.1 Specific comments**

**1.1.1 L4-5: The sentence "On the continental shelves..." is hard to follow**

The sentence was extended to be easier to understand:

> A combination of processes on the continental shelves of the southern Weddell Sea produces the world ocean's densest water (Foldvik et al., 2004). The most important processes are marine heat loss to the atmosphere during sea ice formation and melting of ice shelves from below.

**1.1.2 L65-66, Table 1, and L83:**

> *Which months were the observations conducted? The observation months should be specified in the main text for readers who want know what the "background mean" stands for. (I found the sentence "all CTD measurements were collected in the same season of austral summer" in L325. I think this information should be mentioned earlier.)*

The previous description of "all CTD measurements were collected in the same season of austral summer" was badly placed in the text and a bit too simplistic. We added the following new clarification to the Data section, where the CTD data set is first introduced:

> Due to the prevalent sea ice conditions of the region, the CTD profiles are not evenly distributed across the year, but strongly biased to the austral summer, with 143 out of 168 profiles measured between November and April.

A description of the seasonal distribution is also added to the caption of Table 1:

> The seasonal distribution of the CTD profiles is biased towards the austral summer, with 143 out of 168 profiles measured between November and April.

The exact distribution of CTD profiles across the year can be taken from the stacked histogram of the observation months (subsubsection 1.1.2.

**1.1.3 L178-182: It's hard to follow the methodology because there are vague directives "this" and "it".**

This part of the methodology is rewritten to be more clear:

> Integrating over each tidal peak and summing the resulting energies gives the wave energy at semidiurnal tidal frequencies exceeding the energy of the continuous background. Integrating the energy density over all frequencies $\int_f^N \mathcal{U}(z, \omega) \, d\omega$ gives the total horizontal kinetic energy. The difference of these two energy estimates, the total and the tidal energy, yields the horizontal kinetic energy of the internal wave continuum.

[Figure]

Figure I: Seasonal distribution of the CTD profiles.

**1.1.4 L194: The mooring time series are 2-years long. Does the averaged CTD profiles represent the time-mean density structures?**

Yes, the averaged CTD profiles represent time-mean density structures, as they were taken across multiple years and months. But they still could contain a seasonal bias, as the ship-based measurements are not evenly distributed across the year, but strongly biased to the austral summer, when the low sea ice cover enables expeditions. To estimate how well we can reconstruct time-mean density structures from CTD profiles, we compare them to density structures computed from moored year-round measurements. The required temperature and salinity time series are part of the mooring data sets, but were until now not used. These temperature and salinity time series, at multiple depths on each of the 7 moorings, are first binned to daily averaged values. This choice is made to account for the two different measuring frequencies of the temperature loggers, namely once per hour and six times per hour. Between the upper- and lowermost measurement position of each variable, we can calculate temperature and salinity profiles of low vertical resolution. Where the profiles of both variables vertically and temporally overlap, they could be combined to yield daily density profiles. We added a new paragraph to discuss the potential seasonal bias introduced by the CTD profiles.

> While mooring time series provide year-round data, the seasonal distribution of ship-based CTD profiles is heavily biased towards the austral summer. To see whether the CTD profiles are representative of the long-term mean state, we calculate low-resolution profiles from temperature and salinity measurements from each mooring. Temperature and salinity time series measured at multiple depths on each of the 7 moorings are first binned to daily averaged values and then linearly interpolated in the vertical to yield approximate segments of temperature and salinity. The long-time averages of the segments agree well with mean temperature and salinity profiles calculated from CTD profiles around each mooring. A difference in variability is not observed, as the CTD profiles cover the complete range of values observed by the moored instruments (not shown). Therefore, we can use the CTD profiles to describe the long-term-averaged hydrographic state.

Details can be seen in the comparison figure of mooring- and CTD-derived temperature and salinity profiles (Figure II). To reflect the newly used data for the comparison, we added the new variables to the caption of Table 2 and amended the description:

> From the referenced data sets we use current velocity, in situ temperature, pressure, practical salinity, time, and depth.

During this comparison, we noticed 5 CTD profiles near mooring **C**, **E** and **F** that had escaped our attention in our previous computations. These profiles are far outside the minimum/maximum range of

[Figure]

Figure II: Comparison of mooring- and CTD-derived temperature and salinity profiles.

values measured by the moored instruments and differ strongly from all other CTD profiles (Figure II), even from the same cruise These 5 CTD profiles definitely do not represent a typical stratification at the mooring locations. They are deemed as unphysical due to their deviation and are removed from the dissipation rate calculations. All newly computed results were checked for changes by output regression tests. The results of the Thorpe scales approach, the finestructure parameterization and the IDEMIX parameterization do not change beyond an absolute value of $1 \times 10^{-10}$. The only exception is the IDEMIX parameterization result $\varepsilon_{\text{IGW, IDEMIX}}$ at the instrument position closest to the sea floor at mooring **E**. Here, the estimated wave-induced dissipation rate reduces by $1.1 \times 10^{-9}$. Additionally, due to the removal of the outliers, the observed variability of $N^2$ reduces. Overall, this changes the calculated multiplicative errors $\Delta\varepsilon_{\text{IGW, IDEMIX}}$ everywhere by less than 0.1, except for at the previously mentioned location. There, the multiplicative error reduces from 2.3 to 1.9. The correspondent second-to-last sentence in Appendix C was corrected to reflect the new value of the maximum multiplicative error. All figures showing dissipation rates were corrected to the new versions, but changes are only humanly visible at the named data point. In conclusion, because the corrections are notably affecting only a single data point, our interpretation, and discussion of the results remains the same. The removal of the outliers is now mentioned in the Data section, where the CTD data set is first introduced.

> Five CTD profiles stand out as outliers and are subsequently removed as non-physical profiles from the data analysis, as they differ by many standard deviations from mean background stratification.

**1.1.5 L202-203: I could not understand the sentence "Buoyancy frequency...". Could you rephrase it?**

The sentence was clarified to

> However, internal waves are expected up to a frequency of $N$, which in our case always exceeds the resolved frequencies: time-averaged buoyancy frequencies vary between the different velocity measurement locations, from $8.4\,\text{cpd} \approx 6.1 \times 10^{-4}\,\text{rad s}^{-1}$ to $28.2\,\text{cpd} \approx 2.1 \times 10^{-3}\,\text{rad s}^{-1}$.

**1.1.6 L212: I could not follow the sentence "A second fit ...".**

The sentence was clarified by adding an extra sentence to the previous paragraph "Spectral slope and vertical offset are fitted separately." and rewritten to be now

> In addition, a fit to the resolved part of the total kinetic energy $\mathcal{E}$ determines the vertical offset of the spectral extension.

**1.1.7 L284-285: "This results in ...":**

> *I could not follow this sentence because the authors described that the CTD profiles were depth-binned at 1 or 2 dbar resolution (L68-69). They did not mention the vertical resolution of LADCP profiles.*

The original sentence was about the integration limits for strain variance in the finestructure method. However, these limits are only indirectly dependent on the vertical resolution of the CTD or LADCP profiles. The original sentence was therefore badly placed in the text and too short to be useful. We added to the data section the additional sentence:

> The measured velocity profiles have a vertical resolution of $10\,\mathrm{m}$.

In the text, we moved the sentence about the integration limits used in the strain-based finestructure parameterization inside subsubsection 3.3 to be after equation 8 and added an extra paragraph of explanation.

> We obtain, from each vertical segment of strain, the corresponding strain spectrum in wave number space. By integrating the observed strain spectra $\Phi_{\mathrm{strain}}$, strain variances are determined as [...]

The integration limits used in the shear-based finestructure are moved to the respective section of appendix D, together with an additional explanation.

> Equivalent to the strain-based formulation, shear variances $\langle U_z^2 \rangle$ are computed by integrating shear spectra in wave number space. Because the $10\,\mathrm{m}$ vertical resolution of the shear profiles is lower than the resolution of the strain profiles at 1 to $2\,\mathrm{m}$, less wave numbers are resolved. The integration limits $m_0$ and $m_{\mathrm{c}}$ reflect this difference and cover only over the lowest 8 modes, from $250\,\mathrm{m}$ to $31\,\mathrm{m}$ scales or until the normalized shear variance exceeds a canonical value of 0.66 (Gregg et al., 2003).

In the discussion, we added an extra paragraph about the uncertainty in the finestructure parameterization, connected to the possible choices for the integration limits.

> The results of the finestructure parameterization are influenced by several parameter choices. One example are the integration limits in the variance calculation (see also the discussion in the appendix of Pollmann (2020)). We choose a hybrid approach of confining the upper integration limit $m_{\mathrm{c}}$ with both a fixed minimum length scale and a maximum canonical variance. [...]

During this peer-review, we found our previous integration limits for the shear spectra to be slightly too narrow. The new integration limits of the shear-based finestructure parametrization lead to a moderately improved agreement of shear- and strain-based results. The average shear-to-strain variance ratio $R_w$ value changes from 7.9±10.3 to 7.9±9.8. This therefore do not impact our decision to use the literature value of $R_\omega = 7$ for the strain-based formulation. In summary, because the change to the shear integration limits does not affect the strain-based calculations, our interpretation, and discussion of the results remains the same.

**1.1.8 L319-320: The finestructure parameterization can calculate TKE dissipation rates for all profiles. How did the authors use the background dissipation rates?**

As finestructure parameterization indeed provides complete dissipation rate profiles, any assumption of a background value is not necessary. We assured ourselves, that the actual averaging into bins is done correctly without any false background assumptions (as documented in `scripts/finestructure/finestructure.py` in the software asset). The error was therefore solely in the text, and we are extra grateful to the reviewer for spotting this mistake. We removed the false statement.

**1.1.9 L466-467: "We also observe the largest differences between both wave-induced dissipation rate estimates and the total dissipation." This is not clear from Fig.4.**

The sentence now reads

> Larger differences between $\varepsilon_{\mathrm{total}}$ and $\varepsilon_{\mathrm{IGW}}$ occur in the more turbulent bottom layer (for example, at mooring **B** in Fig. 7).

We revisit the topic in the new Sect. 4.3 *Regional averages of dissipation rates*:

> Even though the result for wave-induced turbulence is not reliable, the large difference between $\langle \varepsilon_{\text{total, Thorpe}} \rangle$ and $\langle \varepsilon_{\text{IGW, fine}} \rangle$ supports the assumption that the bottom layer is largely mixed by processes other than internal wave breaking, like barotropic tides, convection, or friction between mean flow and sea floor.

**1.1.10 L620-621: I could not follow the sentence.**

The sentence was clarified to

> We hypothesize that the variable pattern is caused by spatial variations in the baroclinic tidal wave field.

**1.2 Technical Corrections**

- *L147: (Fig. 2a) → (Fig. 2b)* The correct subfigure Figure 2b is now referenced.

- *L585, RHS of the equation (A1): The constant coefficient should be $\frac{1}{2}$, not 2.* We corrected the numerical factor.

**2    nd reviewers comments**

Comments can be found at `https://egusphere.copernicus.org/preprints/2024/egusphere-2024-2444/#RC2`.

We thank the reviewer for the detailed and constructive assessment of our paper, and are very grateful for the positive feedback regarding the quality of our work.

**2.1    Main comments**

**2.1.1    The significance of the work needs to be better presented.**

To better present the significance of our work, we emphasized our contribution to regional oceanography of the Weddell Sea and to a better understanding of the interaction of gravity currents and internal waves in general. In the introduction, we added:

> Multiple works (Peters and Johns, 2006; Umlauf and Arneborg, 2009; Seim and Fer, 2011; Schaffer et al., 2016) conclude that wave-induced turbulence may be important for entrainment into gravity currents, but without further quantitative analysis of wave contribution. We hence aim to evaluate and quantify the importance of wave-induced turbulence for the WSBW gravity current.

In the conclusion, we refer back to it with

> The potentially crucial role of internal waves as turbulence drivers within gravity currents has been hypothesized or inferred in previous studies, but rarely quantified. Here, we present an approach to estimate wave-induced dissipation rates in a gravity current using standard oceanographic data.

and

> The large scale of the Weddell Sea Bottom Water gravity current makes it a key player in global overturning circulation, but hinders observing its dynamics at high resolution. Our description of spatial distribution and drivers of turbulence contributes to a better understanding of the gravity current and its features.

Additionally, we hpefully better highlight the novelty of our approach of estimating wave-induced dissipation rates from moored velocity time series.

**2.1.2    Fig 6: Method comparison**

> *To compare the three methods, the vertical mean profiles in Fig6 are useful, but they only show the data for part of the transect and misses half of the gravity current it seems (text is a bit unclear as to what is included in the data for Fig 6; see other individual comments). It would be useful to have a similar figure showing the mean vertical mixing profiles of all the data, including closer to the shelf past 51.5°W. If you want to separate mixing estimates within and outside the gravity current (because the finestructure param doesn't work inside the gravity current), then show two mean vertical profile figures.*

Figure 7 (previously Figure 6) now shows 3 examples instead of a single horizontal average:

> Figure 7 shows the results of all methods on the continental shelf at mooring **A**, in the main core of the gravity current at mooring **B** and towards the deeper parts of the gravity current at mooring **E**.

The bottom layer is now hatched in all figures showing finestructure results to mark the values as unreliable. The discussion in the paper about the validity of the methods to estimate wave-induced dissipation rates in the gravity current is extended.

**2.1.3    Assumptions of the finescale parameterizations**

> *If you know that in the homogenously mixed BL, the assumptions for the finescale para are violated, why still present the mixing estimates from that method for that layer in some of the figures?*

In short, we still present the results of the finescale method in the BL, as their ratio to the total dissipation rate informs us about the locally dominating physical processes. But we mark them as unreliable, either by hatching the BL in the figures or by parentheses in Table 3. In Sect. 5.1.2 *Uncertainties of wave-induced dissipation rates* we further discuss the necessary assumptions for estimating $\varepsilon_{\mathrm{IGW}}$ and their validity.

**2.1.4   Great to see you didn't estimate diffusivity**

*Great to see you didn't estimate diffusivity and thanks for adding the small discussion you provide on this Line 471-474*

Thank you for your support of our reasoning.

**2.1.5   Limitations and resolution of the Thorpe scale method**

*There are some ways to use your own dataset to work out if it's the sampling resolution (vertical sampling of CTD) or the instrument accuracy and noise level (CTD resolution) that limit the resolvable density inversions. The parameter R can represent this (Stansfield et al., 2001 and Johnson and Garrett, 2004). Comparing your LT data with a Gaussian fit can also help you estimate how much of the Thorpe scales you have resolved in your data set. This is because the distribution of LT is expected to be lognormal since it is theoretically the result of a multiplicative series of independent events (Stansfield et al., 2001).*

[Figure]

Figure III: Histogram of the measured Thorpe scales along the transect. A log-normal distribution is fitted to it.

We looked more into the limitations and resolution of the Thorpe scale. We added to the Data section the following sentence:

The CTD measurements are accurate to $0.002\,^\circ$C in temperature and $0.002\,\mathrm{g\,kg^{-1}}$ in salinity, which results in a density resolution of similar magnitude of $10^{-3}\,\mathrm{kg\,m^{-3}}$.

We added the results seen in Figure III to the discussion.

We investigate Thorpe scale estimates across the transect by fitting a log-normal distribution to the corresponding histogram (not shown). Although Thorpe scales of few centimeters are physically possible (Johnson and Garrett, 2004), we expect these missing small scales to only contribute little to the overall turbulence pattern, as we resolve the large majority of the theoretically predicted Thorpe scales.

Furthermore, we computed the suggested $R$ parameter (Stansfield et al., 2001) for a selection of density profiles across the slope (Figure IV). The results are presented in the paper:

Across all tested profiles, $R$ stays below a value of 1, confirming density resolution to be the limiting factor (not shown).

[Figure]

Figure IV: Upper row: bin-averaged density profiles of the lowermost 400 m, close to the mooring locations **A**, **B**, and **E**. Cubic density profiles are fitted as a smooth background. Lower row: relative importance parameter $R$ (Stansfield et al., 2001). $R$ is consistently below 1, meaning the results are limited by density resolution.

**2.1.6 Section called 'Connection to larger scales'**

*You start this section by saying that you 'want to set our results in a greater context'. By greater context, I think of the Southern Ocean or global ocean. Restricting that discussion section to the Weddell Sea is not really connecting to 'larger scales' in my mind. I would suggest renaming that section to better reflect the content.*

The section is renamed to "Turbulence along the Weddell Sea Bottom Water gravity current".

**2.1.7 The last paragraph in your Discussion...**

*The last paragraph in your Discussion brings in the topic of climate change and discusses changes in stratification potentially leading to increased vertical mixing. Currently you seem to summarise the findings from Zhou et al. (2023). Are you able to relate better these statements to your findings? Have you tried to see if the mean mixing along the transect has increased between 1989 and 2022? Or are the uncertainties in mixing estimates too high to be able to do that?*

We share the interest of the reviewer to look into potential temporal changes in the dissipation rates over the years. However, mean dissipation rates for each expedition along the transect are not easily comparable, as the expeditions differ in their coverage and resolution of the continental slope. Making sure the resulting time series is as unbiased as possible exceeds the scope of this work. The temporal changes of dissipation rate estimates (longer trends, interannual and seasonal variability) is part of our ongoing work and will be (hopefully) dealt with in a follow-up paper.

**2.2 Specific comments**

**2.2.1 Line 34: You might want to remove the word 'vertical' since all types of mixing, not just vertical, will entrain ambient waters.**

We agree with the reviewer and removed "vertical" from the sentence.

**2.2.2 Line 104-105: Add relevant references such as Dillon 1982, Crawford 1986 and Ferron et al., 1998.**

The connection between Ozmidov and Thorpe scales is now referenced better, following the reviewer's suggestions.

The Thorpe length scale $L_\text{T}$ is linearly related to the Ozmidov scale $L_\text{O}$, at which buoyancy becomes important for eddies (Dillon, 1982; Crawford, 1986; Ferron et al., 1998).

**2.2.3  Line 118: How did you estimate your density noise level**

*Line 118: How did you estimate your density noise level of 3x10-4 kgm-3? Have you considered applying a minimum thickness test based on the Galbraith and Kelley (1996) definition which puts a limit on the resolution of the data set? This minimum height of a density overturn is defined partly on the density accuracy of the instrument.*

The CTD measurements are accurate to $0.002\,^\circ$C in temperature and $0.002\,\text{g kg}^{-1}$ in salinity, which results in a density resolution of similar magnitude of $10^{-3}\,\text{kg m}^{-3}$. The parameter $R$ (Stansfield et al., 2001) shows that this resolution limits our results (Figure IV). For the computation of the Thorpe scales, we previously used an estimated value for the density noise of $3 \times 10^{-4}\,\text{kg m}^{-3}$. With a density resolution of $\mathcal{O}(10^{-3}\,\text{kg m}^{-3})$, we see this now as slightly too low. We increased it to $5 \times 10^{-4}\,\text{kg m}^{-3}$, but still below the density differences we can accurately resolve. This results in that previously accepted overturns, yielding in low dissipation rates of around $3 \times 10^{-10}\,\text{W kg}^{-1}$, are now reclassified as spurious and replaced by the background dissipation of $1 \times 10^{-10}\,\text{W kg}^{-1}$. This especially happens in the open water column towards the east of the transect. The numerical values given in the text are corrected to describe the updated results. The interpretation of the Thorpe scale dissipation rates remains unchanged, as the values change only minimally.

To generally assure us and the readers that only physical overturns are accepted, we use, additionally to the noise criterion, a test suggested by Gargett and Garner (2008): any overturn where the ratio of the vertical distances above and below its inflection point is below 0.2 is rejected as non-physical. We argue that the additional application of the Galbraith and Kelly test to further describe the limits of the Thorpe scale analysis would exceed the scope of this work.

**2.2.4  Line 129-130: Please add a few references here of other people having successfully applied this technique.**

The beginning of Sect. 3.2 *Wave-induced dissipation rate estimates from squared wave energy* now reads

Internal wave energy levels are calculated from moored horizontal velocity time series $u$ and $v$, based on spectral methods. This approach is comparable to previous works on internal waves and their energy (van Haren et al., 2002; Polzin and Lvov, 2011; Le Boyer and Alford, 2021).

**2.2.5  Line 136: Is this P=10 value similar to what is usually applied?**

The general problem persists that although multitaper is not a trivial method, its defining parameters are regularly not given in full in the main text of published literature, while the corresponding research code is not openly accessible. To increase reproducibility without requiring a detailed look into our published research code, we now give all parameters of our multitaper analysis in an additional paragraph in the methods section:

Rotary spectra are calculated from complex velocity time series using the multitaper method (Thomson, 1982; Prieto, 2022). This method repeats spectral calculations of the complex time series in tapered windows and is controlled by three parameters: the time-half-bandwidth product $P$, the number of slepian tapers $k$ and the window width. The time-half-bandwidth product $P$ is usually called $NW$ in literature to reflect its factors, but is here renamed to avoid doubling of variable names. Because the time-half-bandwidth product effectively means that frequencies inside a window of $2P$ Fourier coefficients are smoothed, we chose a value of time-half-bandwidth product $P = 10$ to balance wanted frequency resolution and noise reduction. We use $k = 2P - 1 = 19$ slepian tapers and a window width of the full length of the velocity time series of order of $5 \times 10^3$ points.

While non-oceanographic literature (Thomson, 1982; Cokelaer and Hasch, 2017) recommend $NW$ values between 2.5 and 4 (with a corresponding choice of $2NW$ or $2NW - 1$ slepian tapers), the applications to marine data we found use more varied parameter values. Le Boyer and Alford (2021) use for their multitaper analysis a "window length [...] chosen to be the integer number of inertial periods nearest to 30 days", together with $k = 3$ slepian tapers. They do not give values for the chosen half-bandwidth or

time-half-bandwidth product. If we applied this condition to our measurements, the inertial period at $64°$ S of 13.33 hours would lead to a window length $N$ of 30 days / 13.33 hours $\approx$ 54 data points, very different from our current choices.

However,

> Our choices resemble closely the parameters Chave et al. (2019) use to resolve infragravity waves and tidal frequencies in deep ocean pressure records.

**2.2.6 Line 154-155: add reference to Fig 2b**

Subfigure 3b (previously 2b) is now correctly referenced.

**2.2.7 Line 315-316: What do you base this statement on? 'Luckily, these observed higher modes contain the energy that is dissipated locally through turbulence.' How do you know this?**

The notion that higher vertical modes more likely lead to locally dissipated energy, while lower modes are more likely to spatially transport energy is used several times throughout the paper. This is first mentioned in Sect. 3.2.1 *Wave energy available for local dissipation* and was extended there:

> Wave–wave interactions (for example parametric subharmonic instabilities (Olbers et al., 2020), wave–topography interactions, or wave–mean-flow interactions (Musgrave et al., 2022) transfer the energy to ever smaller scales, where the likelihood for wave breaking increases (Falahat et al., 2014).

and

> Because of the modal dependence of wave–wave interaction time scales discussed above (e.g. Olbers et al., 2020, Fig. 13), it is mostly the high-mode energy that contributes to local turbulence (see also the introduction of de Lavergne et al. (2019), and references therein).

In the criticised part of the text, we now refer back (and link) to this reasoning.

**2.2.8 Line 323-324: Maybe a little more discussion around that choice of neutral density = 28.40 for the gravity current definition is needed: is this a common definition used by more than Naveira et al 2002b?**

The definition of Weddell Sea Bottom Water is not completely unanimous, as two definitions still coexist: as bottom-near water below a certain potential temperature (Foster and Carmack, 1976; Orsi et al., 1999; Nicholls et al., 2009), most recently $< -0.7\,°$C (Vernet et al., 2019; Gordon et al., 2020), or water of neutral density $\gamma^n > 28.40\,\mathrm{kg\,m^{-3}}$. We extend the definition of Weddell Sea Bottom Water in the text with more references:

> We use here the framework of neutral density (Jackett and McDougall, 1997) and define WSBW as water of neutral density $\gamma^n > 28.40\,\mathrm{kg\,m^{-3}}$ (Naveira Garabato et al., 2002; Meredith et al., 2008; Dotto et al., 2014; Llanillo et al., 2023), because it automatically excludes very cold surface waters (Fig. 2).

**2.2.9 Missing "Not shown"**

- *Line 325: if you don't show the approx. 100 m variation in a table or figure, please add 'not shown'.*

- *Line 338-342: If this is not shown in a table or figure, please add 'not shown'*

In both cases, "(not shown)" was added to the text.

**2.2.10 Line 345-346: Outlier profile**

> *That single outlier profile looks dubious. Have you got any other CTD data to check the buoyancy frequency profile from another source? Would there be any reason for such large values of dissipation at that place and that time, like increased wind forcing (storm) or something else?*

Because the outlier in the dissipation rates from the Thorpe scale approach is measured in depths around 3000 m deep in the ocean, it is unlikely that wind forcing could be a physical cause of this. Additionally, profiles from the same expedition do not show segments of dissipation rate this strongly enhanced. The outlier was traced back to a single diagnosed overturn of multiple hundred meter lengths, which was not automatically rejected by the internal quality control in the Thorpe scale algorithm. We removed the large overturn as non-physical, but kept the measurements from the same profile closer to the seafloor. In the text, we added a sentence documenting the outlier removal.

> We manually discard a single overturn (diagnosed around 48° W, ending 200 m above the sea floor), as it is multiple hundreds of meters in length, leading to unrealistic high dissipation rates.

**2.2.11 Line 358: 'around 52°W'**

> *The elevated mixing in the whole water column is at 53oW on Figure 4. Please either change the value in the text or fix the figure.*

We changed the text to be now

> Westward of 52° W, turbulence is elevated throughout the water column, with the highest turbulence observed at the shelf break, around 52.5° W (Fig. 5).

The shift of the bin with the highest observed dissipation rate comes from a redefinition of the easternmost bin edge, as previously the bins of the finestructure and Thorpe scale method were accidentally slightly misaligned.

**2.2.12 Line 375-376: How did you estimate variables like the dissipation rate, inside the gravity current?**

> *Did you do it qualitatively 'by eye' on the figures or did you quantitatively average values within the core of the current based on a core definition? I suggest you try doing the quantitative approach.*

The dissipation rate in each region description was previously determined by averaging qualitatively 'by eye'. In the revised version, we divide the transect into 4 regions (shelf, interfacial layer, bottom layer, open ocean) and calculate an arithmetic mean for each. The definitions and the region-averaged dissipation rates are now presented in Sect. 4.3 *Regional averages of dissipation rates*, with the values given in a new Table 3. In short, the new results confirm the previous qualitative description but allow for an easier comparison between regions and energy sources.

**2.2.13 Line 388-389: Did you only use data between 48.5 and 51.5°W for Figure 6?**

> *I think that is what you mean by this sentence. If so, please add that info in Fig 6 caption.*

This criticism is not applicable any more after the rework of Fig. 6 to the new Fig. 7. We now show three exemplary profiles at moorings **A**, **B**, and **E** instead of a one horizontal average.

**2.2.14 Line 400, 404, 469: add a depth range**

- *Line 400: add a depth range for what you mean by intermittent layer in brackets please.*

- *Line 404: same as above but for 'interfacial layer'; pls add a depth range.*

- *Line 469: Add a depth range for what you call the 'inner water column' please.*

The two layers and regions are now quantitatively defined. At the start of the results section 4.1, we introduce the bottom layer (BL) and the interfacial layer (IL):

> To quantify the two layers, we follow Fer et al. (2010) and define the BL height as the height, where the difference in neutral density to the bottom-most value exceeds $0.01 \, \text{kg m}^{-3}$. The IL above is the region from the edge of the BL to the $20.40 \, \text{kg m}^{-3}$ isopycnal, which defines the upper extent of the gravity current.

In Sect. 4.3 *Regional averages of dissipation rates* we continue with:

> We define the continental shelf region as everything west of $52°\,\text{W}$, where no Weddell Sea Bottom Water is observed. This corresponds to depths shallower than $1000\,\text{m}$ (Fig. 1c, Fig. 2). The open ocean region is then the area above the gravity current, east of $52°\,\text{W}$.

The term "intermittent layer" was a typo and is corrected to interfacial layer.

**2.2.15   Line 464: This has been observed before by Waterman et al. 2014**

> *Add ref and discussion, with here or in your 5.3 'relation to other studies' section, based on existing literature on this topic such as Waterman, S., K. L. Polzin, A. C. Naveira Garabato, K. L. Sheen, and A. Forryan, 2014: Suppression of Internal Wave Breaking in the Antarctic Circumpolar Current near Topography. J. Phys. Oceanogr., 44, 1466–1492, https://doi.org/10.1175/JPO-D-12-0154.1.*

We thank the reviewer for the recommendation of Waterman et al. (2014), which we missed in our literature research. The critiqued original sentence is however about the comparison of both methods for estimating wave-induced dissipation rates. Here, we expect the phenomenon described by Waterman et al. to affect both methods equally. Only in the comparison of total and wave-induced dissipation rates can this bias be observed. We added the following paragraph to the discussion.

> Another indication for a possible overestimation is given by Waterman et al. (2014), who observe overprediction by the finestructure method near topography in the Antarctic Circumpolar Current. They attribute this bias of a factor of 5, compared to microstructure estimations, to not yet understood non-wave mixing processes in the Southern Ocean. This overprediction is observed acting on a vertical scale from the bottom to $1500\,\text{m}$ above the seafloor, far larger than what we consider here. By comparing $\varepsilon_{\text{IGW, fine}}$ to process-blind turbulence estimates $\varepsilon_{\text{total, Thorpe}}$, we see that the finestructure results are consistently lower than the total dissipation rate and therefore physically plausible. Nonetheless, the overprediction described by Waterman et al. (2014) could be a systematic error.

**2.2.16   Line 485-494: This section would benefit from being tidied up.**

> *Currently not very convincing and unclear what you can actually demonstrate based on your data.*

Sect. 5.2 *Wave sources inside and outside the $f$–$N$ frequency range* was rewritten to be more clear. Unfortunately, not much discussed here can be shown here directly from our data. Instead, we rely on reasoning, assumptions, and published previous work.

**2.2.17   Line 525-530: This paragraph is maybe a bit oversimplified?**

> *There are likely some appropriate models that resolve the gravity current and in which the wave propagation would be simulated. Instead of saying it is not possible, maybe say this could be part of future studies when the right tools are identified.*

We rewrote the paragraph to emphasize our requirements on model runs to be able to compare them to our observations. To our knowledge, no existing model run meets these criteria. But this does not mean that it is not possible to simulate wave-induced turbulence in the Weddell Sea Bottom Water gravity current, but a question of the chosen model domain and parameterizations.

**2.2.18   Line 550-558: in this paragraph please better separate your own statements**

> *please better separate your own statements from Zhou et al (2023) findings. Can you better relate what you say here to your own results?*

The paragraph is rewritten to be more clear. While we hypothesize a possible feedback loop, caused by the density loss, it is not certain how much a change in stratification would change the turbulent dynamics.

**2.2.19 Line 552-553: 'The parameterization yields results comparable in value to the long-tested method of finestructure analysis.'**

*This is mostly true but not completely. In the bottom layer, the IDEMIX epsi estimates and the finescale epsi estimates differ significantly in my opinion (compared to the rest of the water column), and in a way that is currently unexplained. I would suggest to temper that statement.*

We further investigated the ratio of the 2 methods for estimating wave-induced dissipation rates (Figure V). In Sect. 4.2 we add:

In the direct comparison, 14 out of the 17 $\varepsilon_{\text{IGW, IDEMIX}}$ estimations are within a factor of 5 to the nearest $\varepsilon_{\text{IGW, fine}}$ result. 11 data points are within a factor of 3 (not shown).

Based on this, we write in the conclusion

The resulting estimates agree reasonably well with those from the established finestructure analysis method, differing in 11 out of 17 data points by less than a factor of 3. Further statistical comparisons are limited by the number of moored velocity time series.

[Figure]

Figure V: Ratio of $\varepsilon_{\text{IGW, fine}}$ to $\varepsilon_{\text{IGW, IDEMIX}}$

**2.3   Technical corrections and Figures**

- *Line 570: add '... is complicated by large uncertainties in the mixing estimates, ...'*
  We extended the sentence according to the reviewer's suggestion.

- *Acronyms throughout: Please define acronyms the first time they are used.*
  We defined the meaning of the acronyms CTD, LADCP, RCM, and IGW at their first occurrence.

- *Line 28: there is something missing in that sentence, like a word and it does not make sense. Please fix that sentence.*
  We rewrote the sentence, together with the added explanation about the WSBW definition.

- *Line 51-52: Consider rephrasing the beginning of that sentence, which is currently awkward 'Due to its remote and difficult to access location at high latitudes, ...'.*
  The order of the clauses in the sentence was switched to be more clear.

- *Line 719: Here and elsewhere in the references, the hyperlinks to the datasets on Pangaea currently include a comma (',') at the end of the link, which makes the link invalid when you click on it. Please remove the comma from within the link so the links can be used to access the data.*

We are unfortunately unable to reproduce the reviewer's problem with the hyperlinks to the datasets. The hyperlinks in our local previously submitted version, in the downloaded preprint and in the current version of the paper resolve correctly for us. If the issue persists, could we get more details on how exactly the bug arises?

- *Figure 1: Very nice. Subplot a and b would benefit from being bigger. Currently it is difficult to look at features on figure 1a as it is too small.*

  Fig. 1 and its subplots were split into now Fig. 1a, b, c and Fig. 2. Additionally, all figures are vector graphics to allow for close zooming.

- *Figure 2b insert: What is the second most prominent frequency that is not labelled, after M2?*

  The second most prominent frequency S2 is now labelled and mentioned in the figure caption.

- *Figure 3: Can you remind the reader in the caption what the measurement period is? Is it January 2017 to January 2019?*

  That is correct, we added the reminder to the figure caption: "Absolute velocities are time-averaged over the moored measurement period from January 2017 to January 2019 and linearly interpolated between measurement locations."

- *Figure 4: Nice figure! It would be useful to see a contour of the core of the gravity current based on the mean velocity field shown on Fig 3. Maybe a contour of 0.30 m/s or 0.25 m/s? In the caption, add info about the grey rectangles which probably mean no data available.*

  Velocity contours were added to Fig. 4, 5, 6 and D1. The meaning of the grey rectangles as "no data" was added to the caption

- *Figure 5: Same as above: add a mean velocity contour to show the location of the core of the gravity current. In the caption, add info about the grey rectangles which probably mean no data available.*

  Velocity contours were added to Fig. 4, 5, 6 and D1. The meaning of the grey rectangles as "no data" was added to the caption

- *Figure 6: Nice figure! See main comments for more feedback.* We also answer to this at the main comments above.

**3 Additional corrections**

While working on the aforementioned changes to the script, the authors found some additional minor shortcomings of the text. Some corrections of them are detailed here.

- At the first mention of RV Polarstern in the main text, we added a new reference (Knust, 2017), which gives technical information about the ship and its operation. The referenced report also acts as the quantifiable, official acknowledgement of the research vessel for the acquirement of data.

- A typo was made in the lowermost row of Table 1, detailing the CTD profile data set. The listed number of profiles per expedition sum to 168 and not to 178. The typo is now corrected to the actual total number of profiles used for the analysis. The table is also reorganized and differentiates now between the name of an expedition and its official ID, both stemming from the description of the referenced data sets.

- The mentions of the cruise reports to RV Polarstern expeditions PS103 and PS117 in the data section are now supported by direct references to them (Boebel, 2017, 2019).

- Because of recent changes to the terms of service of the python package manager *anaconda*, we removed the dependency to it from our code. For the new environment, we also changed the multitaper python package from *spectrum* by Cokelaer and Hasch (2017) to *multitaper* by Prieto (2022), as only the latter supports the newest python versions at the time the calculations were done. Because the parameters are kept the same, the change does not affect the results, as assured by a regression test of the computed $\varepsilon_{\text{IGW, IDEMIX}}$ values and their errors. The references to the used software are adjusted accordingly.

- The westernmost bin edge in the finestructure method and the Thorpe scale method were previously not the same. This is now corrected.

- The hyphen in "Garrett-Munk model" was corrected everywhere to an en-dash "Garrett–Munk model".

- In the acknowledgements, we added the grant numbers to the RV Polarstern expeditions led by the HAFOS project: PS103 and PS117 provided the mooring data, which was extensively used in this work. PS129 provided a CTD section and the LADCP data, which was essential for calibrating the finestructure method.

**References**

Boebel, Olaf (Jan. 2017). *The Expedition PS103 of the Research Vessel POLARSTERN to the Weddell Sea in 2016/2017*. en. Tech. rep. Artwork Size: 160 pages ISSN: 1866-3192 Medium: application/pdf Publication Title: Berichte zur Polar- und Meeresforschung = Reports on Polar and Marine Research Version Number: 1.0 Volume: 710. Alfred-Wegener-Institut, Helmholtz-Zentrum für Polar- und Meeresforschung, pp. 1–160. DOI: 10.2312/BZPM_0710_2017. URL: https://www.tib.eu/suchen/id/awi:0e128216d1ec6a93343f78d0d48dd812dda63563 (visited on 08/16/2024).

— (Jan. 2019). *The Expedition PS117 of the Research Vessel POLARSTERN to the Weddell Sea in 2018/2019*. en. Tech. rep. Artwork Size: 205 pages ISSN: 1866-3192 Medium: application/pdf Publication Title: Berichte zur Polar- und Meeresforschung = Reports on Polar and Marine Research Version Number: 1.0 Volume: 732. Alfred-Wegener-Institut, Helmholtz-Zentrum für Polar- und Meeresforschung, pp. 1–205. DOI: 10.2312/BZPM_0732_2019. URL: https://www.tib.eu/suchen/id/awi:35622b700696e9af3f7fc688ab85c009f1dc8a84 (visited on 08/16/2024).

Chave, Alan D. et al. (Mar. 2019). "High- $Q$ Spectral Peaks and Nonstationarity in the Deep Ocean Infragravity Wave Band: Tidal Harmonics and Solar Normal Modes". en. In: *Journal of Geophysical Research: Oceans* 124.3, pp. 2072–2087. ISSN: 2169-9275, 2169-9291. DOI: 10.1029/2018JC014586. URL: https://agupubs.onlinelibrary.wiley.com/doi/10.1029/2018JC014586 (visited on 11/22/2024).

Cokelaer, Thomas and Juergen Hasch (Oct. 2017). "'Spectrum': Spectral Analysis in Python". In: *The Journal of Open Source Software* 2.18, p. 348. ISSN: 2475-9066. DOI: 10.21105/joss.00348. URL: http://joss.theoj.org/papers/10.21105/joss.00348 (visited on 05/28/2024).

Crawford, William R. (Nov. 1986). "A Comparison of Length Scales and Decay Times of Turbulence in Stably Stratified Flows". en. In: *Journal of Physical Oceanography* 16.11, pp. 1847–1854. ISSN: 0022-3670, 1520-0485. DOI: 10.1175/1520-0485(1986)016<1847:ACOLSA>2.0.CO;2. URL: http://journals.ametsoc.org/doi/10.1175/1520-0485(1986)016%3C1847:ACOLSA%3E2.0.CO;2 (visited on 11/18/2024).

De Lavergne, C. et al. (May 2019). "Toward global maps of internal tide energy sinks". In: *Ocean Modelling* 137, pp. 52–75. ISSN: 14635003. DOI: 10.1016/j.ocemod.2019.03.010.

Dillon, T. M. (Nov. 1982). "Vertical overturns: A comparison of Thorpe and Ozmidov length scales". en. In: *Journal of Geophysical Research: Oceans* 87.C12, pp. 9601–9613. ISSN: 0148-0227. DOI: 10.1029/JC087iC12p09601. URL: https://agupubs.onlinelibrary.wiley.com/doi/10.1029/JC087iC12p09601 (visited on 10/16/2023).

Dotto, T. S. et al. (June 2014). "Assessment of the structure and variability of Weddell Sea water masses in distinct ocean reanalysis products". en. In: *Ocean Science* 10.3, pp. 523–546. ISSN: 1812-0792. DOI: 10.5194/os-10-523-2014. URL: https://os.copernicus.org/articles/10/523/2014/ (visited on 11/21/2024).

Falahat, Saeed et al. (Dec. 2014). "Global Calculation of Tidal Energy Conversion into Vertical Normal Modes". en. In: *Journal of Physical Oceanography* 44.12, pp. 3225–3244. ISSN: 0022-3670, 1520-0485. DOI: 10.1175/JPO-D-14-0002.1. URL: http://journals.ametsoc.org/doi/10.1175/JPO-D-14-0002.1 (visited on 04/10/2024).

Fer, Ilker et al. (Jan. 2010). "Intense mixing of the Faroe Bank Channel overflow". en. In: *Geophysical Research Letters* 37.2. ISSN: 00948276. DOI: 10.1029/2009GL041924. URL: http://doi.wiley.com/10.1029/2009GL041924 (visited on 08/22/2023).

Ferron, Bruno et al. (Oct. 1998). "Mixing in the Romanche Fracture Zone". en. In: *Journal of Physical Oceanography* 28.10, pp. 1929–1945. ISSN: 0022-3670, 1520-0485. DOI: 10.1175/1520-0485(1998)028<1929:MITRFZ>2.0.CO;2. URL: http://journals.ametsoc.org/doi/10.1175/1520-0485(1998)028%3C1929:MITRFZ%3E2.0.CO;2 (visited on 02/20/2024).

Foldvik, A. et al. (Feb. 2004). "Ice shelf water overflow and bottom water formation in the southern Weddell Sea". In: *Journal of Geophysical Research: Oceans* 109.2. Publisher: Blackwell Publishing Ltd, p. C02015. ISSN: 21699291. DOI: 10.1029/2003jc002008.

Foster, Theodore D. and Eddy C. Carmack (Apr. 1976). "Frontal zone mixing and Antarctic Bottom water formation in the southern Weddell Sea". en. In: *Deep Sea Research and Oceanographic Abstracts* 23.4, pp. 301–317. ISSN: 00117471. DOI: 10.1016/0011-7471(76)90872-X. URL: https://linkinghub.elsevier.com/retrieve/pii/001174717690872X (visited on 11/21/2024).

Gargett, Ann and Teresa Garner (Sept. 2008). "Determining Thorpe Scales from Ship-Lowered CTD Density Profiles". en. In: *Journal of Atmospheric and Oceanic Technology* 25.9, pp. 1657–1670. ISSN: 1520-0426, 0739-0572. DOI: 10.1175/2008JTECHO541.1. URL: http://journals.ametsoc.org/doi/10.1175/2008JTECHO541.1 (visited on 12/05/2024).

Gordon, Arnold L. et al. (Feb. 2020). "Interannual Variability of the Outflow of Weddell Sea Bottom Water". en. In: *Geophysical Research Letters* 47.4. ISSN: 0094-8276, 1944-8007. DOI: 10.1029/2020GL087014. URL: https://onlinelibrary.wiley.com/doi/10.1029/2020GL087014 (visited on 07/07/2022).

Gregg, Michael C. et al. (Apr. 2003). "Reduced mixing from the breaking of internal waves in equatorial waters". en. In: *Nature* 422.6931, pp. 513–515. ISSN: 0028-0836, 1476-4687. DOI: 10.1038/nature01507. URL: https://www.nature.com/articles/nature01507 (visited on 10/23/2024).

Jackett, David R. and Trevor J. McDougall (Feb. 1997). "A Neutral Density Variable for the World's Oceans". en. In: *Journal of Physical Oceanography* 27.2, pp. 237–263. ISSN: 0022-3670, 1520-0485. DOI: 10.1175/1520-0485(1997)027<0237:ANDVFT>2.0.CO;2. URL: http://journals.ametsoc.org/doi/10.1175/1520-0485(1997)027%3C0237:ANDVFT%3E2.0.CO;2 (visited on 07/06/2024).

Johnson, Helen L. and Chris Garrett (Nov. 2004). "Effects of Noise on Thorpe Scales and Run Lengths". en. In: *Journal of Physical Oceanography* 34.11, pp. 2359–2372. ISSN: 1520-0485, 0022-3670. DOI: 10.1175/JPO2641.1. URL: http://journals.ametsoc.org/doi/10.1175/JPO2641.1 (visited on 11/18/2024).

Knust, Rainer (Oct. 2017). "Polar Research and Supply Vessel POLARSTERN Operated by the Alfred-Wegener-Institute". In: *Journal of large-scale research facilities JLSRF* 3, A119. ISSN: 2364-091X. DOI: 10.17815/jlsrf-3-163. URL: https://jlsrf.org/index.php/lsf/article/view/163 (visited on 08/16/2024).

Le Boyer, Arnaud and Matthew H. Alford (Sept. 2021). "Variability and Sources of the Internal Wave Continuum Examined from Global Moored Velocity Records". en. In: *Journal of Physical Oceanog-*

*raphy* 51.9, pp. 2807–2823. ISSN: 0022-3670, 1520-0485. DOI: 10.1175/JPO-D-20-0155.1. URL: https://journals.ametsoc.org/view/journals/phoc/51/9/JPO-D-20-0155.1.xml (visited on 08/21/2023).

Llanillo, Pedro J. et al. (Feb. 2023). "The Deep-Water Plume in the Northwestern Weddell Sea, Antarctica: Mean State, Seasonal Cycle and Interannual Variability Influenced by Climate Modes". en. In: *Journal of Geophysical Research: Oceans* 128.2, e2022JC019375. ISSN: 2169-9275, 2169-9291. DOI: 10.1029/2022JC019375. URL: https://agupubs.onlinelibrary.wiley.com/doi/10.1029/2022JC019375 (visited on 10/10/2023).

Meredith, Michael P. et al. (July 2008). "Evolution of the Deep and Bottom Waters of the Scotia Sea, Southern Ocean, during 1995–2005*". In: *Journal of Climate* 21.13, pp. 3327–3343. ISSN: 1520-0442, 0894-8755. DOI: 10.1175/2007JCLI2238.1.

Musgrave, Ruth et al. (2022). "The lifecycle of topographically-generated internal waves". en. In: *Ocean Mixing*. Elsevier, pp. 117–144. ISBN: 978-0-12-821512-8. DOI: 10.1016/B978-0-12-821512-8.00013-X. URL: https://linkinghub.elsevier.com/retrieve/pii/B978012821512800013X (visited on 09/11/2023).

Naveira Garabato, Alberto C. et al. (Jan. 2002). "On the export of Antarctic Bottom Water from the Weddell Sea". en. In: *Deep Sea Research Part II: Topical Studies in Oceanography* 49.21, pp. 4715–4742. ISSN: 09670645. DOI: 10.1016/S0967-0645(02)00156-X. URL: https://linkinghub.elsevier.com/retrieve/pii/S096706450200156X (visited on 02/09/2023).

Nicholls, Keith W. et al. (July 2009). "Ice-ocean processes over the continental shelf of the southern Weddell Sea, Antarctica: A review". In: *Reviews of Geophysics* 47.3, RG3003. ISSN: 8755-1209. DOI: 10.1029/2007RG000250.

Olbers, Dirk et al. (Mar. 2020). "On PSI Interactions in Internal Gravity Wave Fields and the Decay of Baroclinic Tides". en. In: *Journal of Physical Oceanography* 50.3, pp. 751–771. ISSN: 0022-3670, 1520-0485. DOI: 10.1175/JPO-D-19-0224.1. URL: https://journals.ametsoc.org/view/journals/phoc/50/3/jpo-d-19-0224.1.xml (visited on 04/16/2024).

Orsi, A.H. et al. (Jan. 1999). "Circulation, mixing, and production of Antarctic Bottom Water". en. In: *Progress in Oceanography* 43.1, pp. 55–109. ISSN: 00796611. DOI: 10.1016/S0079-6611(99)00004-X. URL: https://linkinghub.elsevier.com/retrieve/pii/S007966119900004X (visited on 11/21/2024).

Peters, Hartmut and William E. Johns (Sept. 2006). "Bottom Layer Turbulence in the Red Sea Outflow Plume". en. In: *Journal of Physical Oceanography* 36.9, pp. 1763–1785. ISSN: 1520-0485, 0022-3670. DOI: 10.1175/JPO2939.1. URL: http://journals.ametsoc.org/doi/10.1175/JPO2939.1 (visited on 01/22/2025).

Pollmann, Friederike (June 2020). "Global characterization of the ocean's internal wave spectrum". In: *Journal of Physical Oceanography* 50.7. Publisher: American Meteorological Society, pp. 1871–1891. ISSN: 15200485. DOI: 10.1175/JPO-D-19-0185.1.

Polzin, K. L. and Y. V. Lvov (Nov. 2011). "Toward Regional Characterizations of the Oceanic Internal Wavefield". In: *Reviews of Geophysics* 49.4, RG4003. ISSN: 8755-1209. DOI: 10.1029/2010RG000329.

Prieto, Germán A. (May 2022). "The *Multitaper* Spectrum Analysis Package in Python". en. In: *Seismological Research Letters* 93.3, pp. 1922–1929. ISSN: 0895-0695, 1938-2057. DOI: 10.1785/0220210332. URL: https://pubs.geoscienceworld.org/srl/article/93/3/1922/612834/The-Multitaper-Spectrum-Analysis-Package-in-Python (visited on 02/09/2024).

Schaffer, Janin et al. (Oct. 2016). "Enhanced turbulence driven by mesoscale motions and flow-topography interaction in the Denmark Strait Overflow plume: Enhanced turbulence in the DSO plume". en. In: *Journal of Geophysical Research: Oceans* 121.10, pp. 7650–7672. ISSN: 21699275. DOI: 10.1002/2016JC011653. URL: http://doi.wiley.com/10.1002/2016JC011653 (visited on 07/13/2023).

Seim, Knut S. and Ilker Fer (July 2011). "Mixing in the stratified interface of the Faroe Bank Channel overflow: The role of transverse circulation and internal waves". en. In: *Journal of Geophysical Research: Oceans* 116.C7, 2010JC006805. ISSN: 0148-0227. DOI: 10.1029/2010JC006805. URL: https://agupubs.onlinelibrary.wiley.com/doi/10.1029/2010JC006805 (visited on 06/01/2024).

Stansfield, Kate et al. (Dec. 2001). "The Probability Distribution of the Thorpe Displacement within Overturns in Juan de Fuca Strait". en. In: *Journal of Physical Oceanography* 31.12, pp. 3421–3434. ISSN: 0022-3670, 1520-0485. DOI: 10.1175/1520-0485(2001)031<3421:TPDOTT>2.0.CO;2. URL: http://journals.ametsoc.org/doi/10.1175/1520-0485(2001)031%3C3421:TPDOTT%3E2.0.CO;2 (visited on 11/18/2024).

Thomson, D.J. (1982). "Spectrum estimation and harmonic analysis". en. In: *Proceedings of the IEEE* 70.9, pp. 1055–1096. ISSN: 0018-9219. DOI: 10.1109/PROC.1982.12433. URL: http://ieeexplore.ieee.org/document/1456701/ (visited on 08/15/2024).

Umlauf, Lars and Lars Arneborg (Oct. 2009). "Dynamics of Rotating Shallow Gravity Currents Passing through a Channel. Part I: Observation of Transverse Structure". In: *Journal of Physical Oceanography* 39.10, pp. 2385–2401. ISSN: 1520-0485, 0022-3670. DOI: 10.1175/2009JPO4159.1.

Van Haren, Hans et al. (2002). "On the nature of internal wave spectra near a continental slope". In: *Geophysical Research Letters* 29.12, p. 1615. ISSN: 0094-8276. DOI: 10.1029/2001GL014341.

Vernet, M. et al. (Sept. 2019). "The Weddell Gyre, Southern Ocean: Present Knowledge and Future Challenges". In: *Reviews of Geophysics* 57.3, pp. 623–708. ISSN: 8755-1209, 1944-9208. DOI: 10.1029/2018RG000604.

Waterman, Stephanie et al. (May 2014). "Suppression of Internal Wave Breaking in the Antarctic Circumpolar Current near Topography". en. In: *Journal of Physical Oceanography* 44.5, pp. 1466–1492. ISSN: 0022-3670, 1520-0485. DOI: 10.1175/JPO-D-12-0154.1. URL: http://journals.ametsoc.org/doi/10.1175/JPO-D-12-0154.1 (visited on 11/15/2024).

---

## Author Response (AR2)

**Author's response to the 3rd minor revision**

Ole Pinner (correspondent author)

February 4, 2025

**1 Line 66: Shipboard-velocity data**

We agree with the editor and clarified in the introduction that we use velocity profiles from only a single cruise.

**2 Line 581: Number of hydrographic profiles**

We agree with the editor and remind now the reader about the number of hydrographic profiles which include velocity data.